# Frictional fluid instabilities shaped by viscous forces

Dawang Zhang[1], James M. Campbell[1,2], Jon A. Eriksen[2], Eirik G. Flekkøy[2,3], Knut Jørgen Måløy [2,4], Christopher W. MacMinn [5] & Bjørnar Sandnes [1] ✉

Multiphase flows involving granular materials are complex and prone to pattern formation caused by competing mechanical and hydrodynamic interactions. Here we study the interplay between granular bulldozing and the stabilising effect of viscous pressure gradients in the invading fluid. Injection of aqueous solutions into layers of dry, hydrophobic grains represent a viscously stable scenario where we observe a transition from growth of a single frictional finger to simultaneous growth of multiple fingers as viscous forces are increased. The pattern is made more compact by the internal viscous pressure gradient, ultimately resulting in a fully stabilised front of frictional fingers advancing as a radial spoke pattern.

Viscous multiphase flows involving two fluids and a granular material occur in such diverse scenarios as mud and debris flows, methane venting from sediments, degassing of volatiles from magma, and the processing of granular and particulate systems in the food, pharmaceutical, and chemical industries[1–13]. The presence of the granular material introduces solid friction as a governing force in the dynamics, alongside viscosity, capillarity, and gravity. This multitude of interacting elements and forces can give rise to instabilities and the emergence of patterns[14–31], making these multiphase frictional flows inherently difficult to predict or control.

Multiphase frictional flows inhabit a large parameter space, but relatively few scenarios have attracted any attention. One such flow that is now relatively well understood is the injection of a low-viscosity invading fluid (viscosity $\eta_{\mathrm{inv}}$) to displace a much more viscous defending fluid (viscosity $\eta_{\mathrm{def}} \gg \eta_{\mathrm{inv}}$) containing sedimented grains, for the case where the invading fluid is nonwetting to the grains (i.e. drainage). Without the grains, or with grains that are fixed in place (i.e. within a rigid porous medium), this problem is famously viscously unstable and will be subject to classical viscous fingering (i.e. the Saffman–Taylor instability)[32–35]. With movable grains, the nonwetting invading phase will tend to bulldoze the defending mixture rather than invading the space between or above the grains as long as capillary forces are strong enough to overcome friction with the wall(s) and among the grains (i.e. the capillary entry pressure must be sufficiently larger than the frictional resistance to sliding and rearrangement). This bulldozing behaviour further destabilises the system as the accumulation of grains on the defending side of the interface penalises uniform displacement, leading to the formation of fractures, fingers, bubbles, labyrinths, and other patterns, depending on the injection rate and the packing fraction[15,19,28].

Without the grains, reversing the two viscosities ($\eta_{\mathrm{def}} < \eta_{\mathrm{inv}}$) negates the fingering instability by turning viscous pressure into a stabilising mechanism. With grains that are fixed in place, capillary forces and pore-scale disorder compete with viscous stabilisation to produce fractal invasion-percolation patterns at low injection rates and rough but stable fronts at high injection rates[34–38]. With movable grains, however, the flow is frictionally unstable at all rates due to bulldozing. The competition between these two mechanisms has not previously been studied in any detail and even basic questions remain unanswered; for example, to what extent can viscous forces stabilise the flow against the frictional instability? Here, we explore this competition systematically using experiments and simulations. Focusing on the case where $\eta_{\mathrm{inv}} \gg \eta_{\mathrm{def}}$, we show that the pattern formation is controlled by the strength of viscous forces in the invading phase relative to friction due to bulldozing and pile-up of grains in the defending phase, as quantified by a "viscous deformability" parameter $D_{\mathrm{visc}}$. Increasing $D_{\mathrm{visc}}$ leads to a striking transition from the growth of one solitary finger to the simultaneous growth of multiple, wandering fingers to the axisymmetric growth of a radial spoke pattern as the flow is increasingly viscously stabilised.

[1]Department of Chemical Engineering, Swansea University, Swansea SA1 8EN, UK. [2]PoreLab, Njord Center, Department of Physics, University of Oslo, N-0371 Oslo, Norway. [3]PoreLab, Department of Chemistry, Norwegian University of Science and Technology, N-7491 Trondheim, Norway. [4]PoreLab, Department of Geoscience and Petroleum, Norwegian University of Science and Technology, N-7491 Trondheim, Norway. [5]Department of Engineering Science, University of Oxford, Oxford OX1 3PJ, UK. ✉e-mail: b.sandnes@swansea.ac.uk

## Results

### Viscously stable frictional fingers

Viscously stable frictional fingering was achieved by injecting water or a mixture of water and glycerol at a flow rate $Q$ into a Hele-Shaw cell comprising two parallel glass plates separated by a gap of thickness $b = 0.9$ mm and containing a dry layer of polydisperse hydrophobic beads (Fig. 1a). As such, air was the low viscosity defending fluid in which the grains were "submerged". We define the log viscosity ratio $\mathcal{M} = \log(\eta_{def}/\eta_{inv})$, which is negative for viscously stable scenarios and strongly negative here, where $\eta_{def} \ll \eta_{inv}$. The flow cells were prepared with various filling levels $\varphi = h/b$ of beads, where $h$ is the initial thickness of the layer; the beads were silanised glass, sieved between 70 and 100 $\mu$m, with mean diameter $d = 87\,\mu$m.

In all cases, the frictional instability shaped the invading water into one or more fingers of width $2R$ surrounded by a compaction front of thickness $L$ of dry, bulldozed grains (Fig. 1b and c). These fingers grew only at their tips; the side-walls were immobile after their initial formation, except where new fingers were initiated at larger values of $Q$, as discussed below.

Figure 2a shows a selection of experimental results for different $Q$ and $\varphi$. Figure 2b shows corresponding simulations described in a later section. Low injection rates resulted in the growth of a single finger that "wormed" its way through the cell (e.g. $Q = 1$ ml/min, bottom row in Fig. 2, Supplementary Movie 1). At the same injection rate, increasing $\varphi$ led to a narrower finger because the compaction front thickened

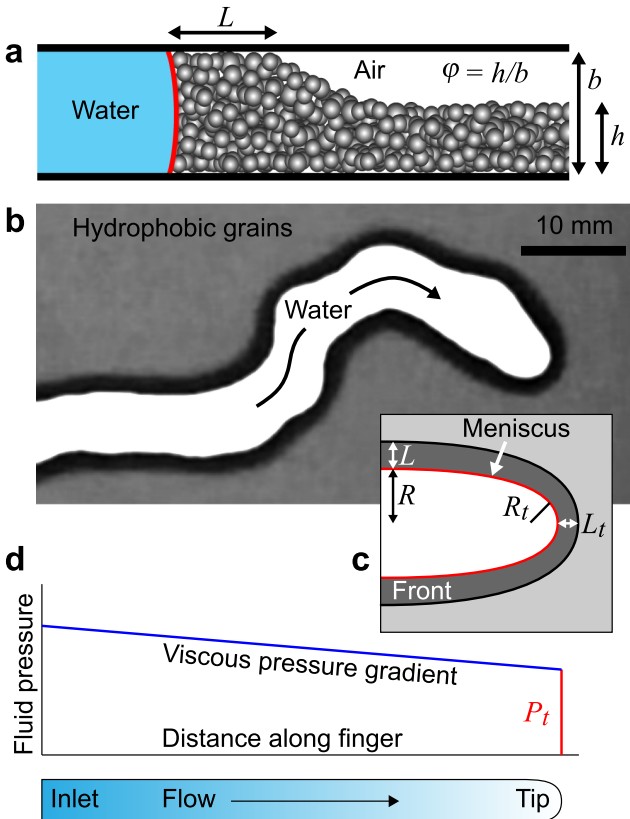

**Fig. 1 | Frictional fingering in hydrophobic grains. a** Schematic cross-section of an invading fluid-fluid interface (red line) bulldozing a layer of hydrophobic grains with initial filling level $\varphi$ into a compaction front of width $L$ that bridges the gap between the plates. **b** Photographic top view of an invading finger. The white region has been invaded with water and cleared of grains, while the black region (the compaction front) has been completely filled with grains. **c** Advancing finger tips have radius of curvature $R_t$ and compaction-front width $L_t$; away from the tips, fully expanded fingers have half-width $R$ and compaction-front width $L$. **d** Viscous flow of liquid along a growing finger leads to a pressure gradient from the injection pressure at the inlet to the capillary pressure $P_t$ at the tip.

more rapidly with the advance of the interface (Fig. 3)[15,39]. Values for $R$ were obtained from final images (such as those presented in Fig. 2) by measuring the ratio of fluid-filled invaded area to internal finger interface length $R = A_{fluid}/S_{finger}$.

The grains in the compaction front bridge the gap between the plates. In a straight segment of the side-wall, the capillary pressure imparted by the meniscus is opposed by the effective stress in the granular material, which disperses through grain-grain contacts to the plates. Deformation of the front requires dilation, which is opposed by the confining plates and the granular pressure from neighbouring front segments. Assuming Coulomb friction between the granular material and the plates and that out-of-plane stresses are proportional to the imposed in-plane stress (the "Janssen law"[40]), the frictional stress resisting motion of the front increases exponentially with front width: $\sigma(L) \propto e^{L/\xi}$, where $\xi = b/(2\mu\kappa)$ is the characteristic length, $\mu$ is the effective coefficient of friction between the grains and the plates, and $\kappa$ is the ratio of out-of-plane normal stress to in-plane normal stress in the granular packing (i.e. the Janssen coefficient)[39,41].

At the tip of a growing finger, the front is curved (Fig. 1c) and the streamlines of granular motion diverge as the grains are pushed outwards normal to the interface. In the reference frame of the moving tip, the granular material in the front is therefore continuously being stretched tangentially to the interface. This extensional motion reduces the granular pressure against the plates by weakening tangential granular force chains. The confinement-induced jamming that produced exponentially increasing friction at the straight side-walls is therefore significantly reduced at the moving tip, where the finger width is set. In agreement with previous work[42], we find that a simple linear friction model provides a good fit to the experimental data in Fig. 3 (see model results described below). Following[42], we estimate the frictional stress at the tip as $\sigma_t \approx \sigma_0 L_t/b$, where $L_t$ is the front width at the tip (see Fig. 1c) and $\sigma_0$ represents the total frictional force per unit contact area with the plates; we use $\sigma_0$ as a fitting constant.

At the curved tip, the menisci in the pores between the grains collectively generate an effective interfacial tension at the scale of several grains that acts to minimise the curvature of the tip. The fluid pressure at the tip of a growing finger is therefore partly opposed by the Laplace pressure $\gamma_{eff}/R_t$ associated with its in-plane curvature $R_t$ (Fig. 1), where $\gamma_{eff}$ is the effective interfacial tension at the liquid–gas/ grain interface[39]. We approximate the latter with the liquid–gas interfacial tension, $\gamma_{eff} \approx \gamma$. Note that we ignore the out-of-plane curvature because it is independent of the finger shape.

Following previous work[15,42,43], we now estimate the characteristic rate-independent finger width $2R_f$ that balances capillarity with friction by seeking the value of $R_f$ that minimises the capillary pressure at the tip $P_t$. Conservation of mass for the finger as a whole suggests that $(1 - \varphi)L_f \approx \varphi R_f$, where $L_f$ is the characteristic front width and where we have neglected the small tip region; we assume for simplicity that the same argument applies independently at the tip, $(1 - \varphi)L_t \approx \varphi R_t$. From the arguments above, the total yield pressure at the fingertip is then $P_t \approx \sigma_0 L_t/b + \gamma/R_t \approx (\sigma_0/b)[\varphi/(1 - \varphi)]R_t + \gamma/R_t$, suggesting that friction favours narrower fingers (displacing fewer grains) while surface tension favours wider fingers (less curvature). We then identify the $R_t$ at which $P_t$ is minimised by setting $dP_t/dR_t = 0$. Finally, we link $R_t$ to $R_f$ by requiring that this same yield pressure $P_t$ must apply along the straight side walls of the finger where the frictional stress is $\sigma_0 L_f/b$ and the curvature is negligible, noting that we are neglecting viscosity over the distance $\sim 2R_f$ over which fingers transition from the curved tip to the straight side walls. Combining these ingredients, we arrive at a simple expression for $R_f$:

$$R_f = 2\sqrt{\frac{\gamma b(1 - \varphi)}{\sigma_0 \varphi}}. \tag{1}$$

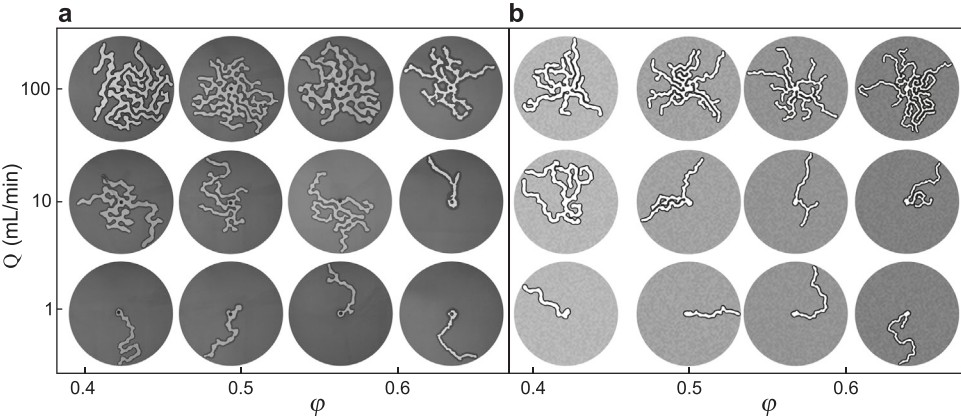

**Fig. 2 | Phase diagram of water invasion patterns at different $\varphi$ and Q.**
**a** Experiments and (**b**) simulations. Water (bright fingering structure) invades from a central inlet, displacing a layer of dry hydrophobic beads (dark grey) (see Supplementary Movies 1 and 2). All images show the moment when the first finger reaches a radius of $r_{out} = 13.4$ cm and are cropped to that radius.

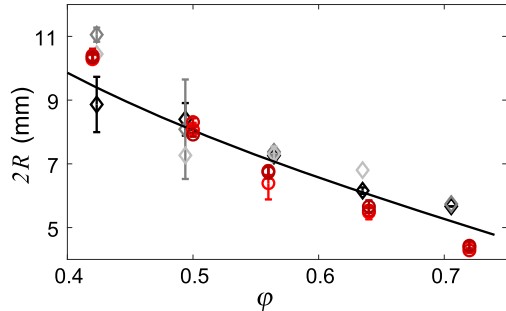

**Fig. 3 | Frictional finger width 2R decreases with filling fraction $\varphi$.** Diamonds and circles show experimental and simulation results, respectively (mean and standard deviation of three repeats). Black, dark grey and light grey represent experiments at $Q = 1, 3$ and 10 mL/min respectively. Red circles represent simulation data at $Q = 1, 3$ and 10 mL/min. $R$ is measured as the ratio of total invaded area to total interface length. The solid line shows the characteristic finger width $2R_f(\varphi)$ from Eq. (1), having taken $\sigma_0 = 16$ Pa as the best fit to the experimental results.

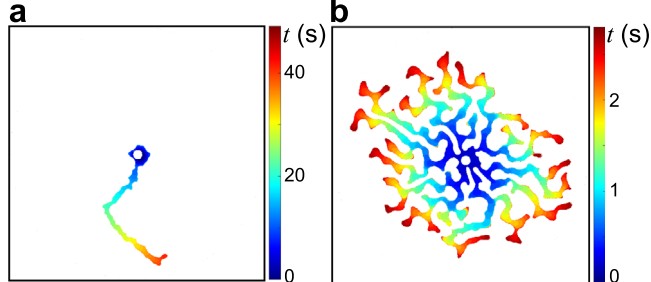

**Fig. 4 | Transition from a single finger to multiple fingers as Q increases. a** Time evolution of a single finger ($Q = 1$ mL/min) and (**b**) multiple fingers ($Q = 200$ mL/min) colourised according to invasion time $t$.

This finger width $2R_f$ is thus the emergent natural length scale in our system. A very similar expression has been derived and used for viscously unstable frictional fingers[15,42,43].

We plot $2R$ as a function of $\varphi$ in Fig. 3 for experiments and simulations at low rates $Q$, comparing against $2R_f(\varphi)$ from Eq. (1). The theory agrees well with the experimental observations, in common with previous research on viscously unstable air-invasion labyrinths[15,42] and confirming the expectation that there exists a rate-independent regime where viscous forces within the fingers are indeed negligible, such that capillarity and friction compete to set the finger width.

**Transition from a single finger to multiple fingers**
As we increase $Q$ over two orders of magnitude, we observe a transition toward a regime where viscous pressure gradients become significant. As $Q$ increases, the number of simultaneously growing fingers increases monotonically from one at low $Q$ to nearly 20 at the highest values of $Q$ explored here (Figs. 2 and 4, Supplementary Movie 2). Thus, strikingly, viscous forces drive the pattern to be more space-filling in this system by promoting the formation of more fingers.

Low- and high-$Q$ experiments are compared in Fig. 4a and b, where the invading-fluid-filled fingers are colourised by invasion time. At low $Q$, growth is localised to the tip of a single finger into which all the injected fluid flows. Branching is sometimes observed, but usually only one finger is actively growing at any time. At high $Q$, several fingers grow simultaneously, as indicated by the rough axisymmetry of the colours in Fig. 4b. New fingers sprout by side-branching as the injected fluid flows through the network of fingers toward the active tips.

The viscous pressure gradient is located within the viscous invading fluid, with the pressure decreasing along the fingers from the inlet toward the moving tips (Fig. 1d). A sufficiently high pressure along the length of a finger can drive new fingers to break out from the side walls, leading to growth in the central parts of the pattern; this is the manifestation of viscous stabilisation in this frictionally unstable system. Figure 5a shows that the number of active fingers $N$ increases with injected volume $V$ and with injection rate $Q$, which is consistent with the fact that the fluid pressure at any fixed radial position along a finger must increase with the length of that finger and with the flow rate through it.

Breakout of new fingers is suppressed by at least two mechanisms. First, the stress required to deform a straight side-wall is higher than at the already curved tip, where the divergent flow of grains relieves bridging stresses by accommodating dilation. Second, the coefficient of static friction $\mu_s$ at the side walls is likely to be higher than the coefficient of dynamic friction $\mu_d$ at the moving tips, leading to higher frictional strength along the side walls. We represent these effects by introducing a higher threshold pressure $P_b = P_t + \Delta P_b$ required to sprout a new finger from a side wall.

The results in §II A suggest that larger values of $\varphi$ produce thicker compaction fronts, $L_f \propto \sqrt{\varphi/(1 - \varphi)}$[15]. These thicker fronts should provide a stronger resistance to breakout, meaning that $\Delta P_b$ should increase with $L_f$; thus, it should become more difficult to form new

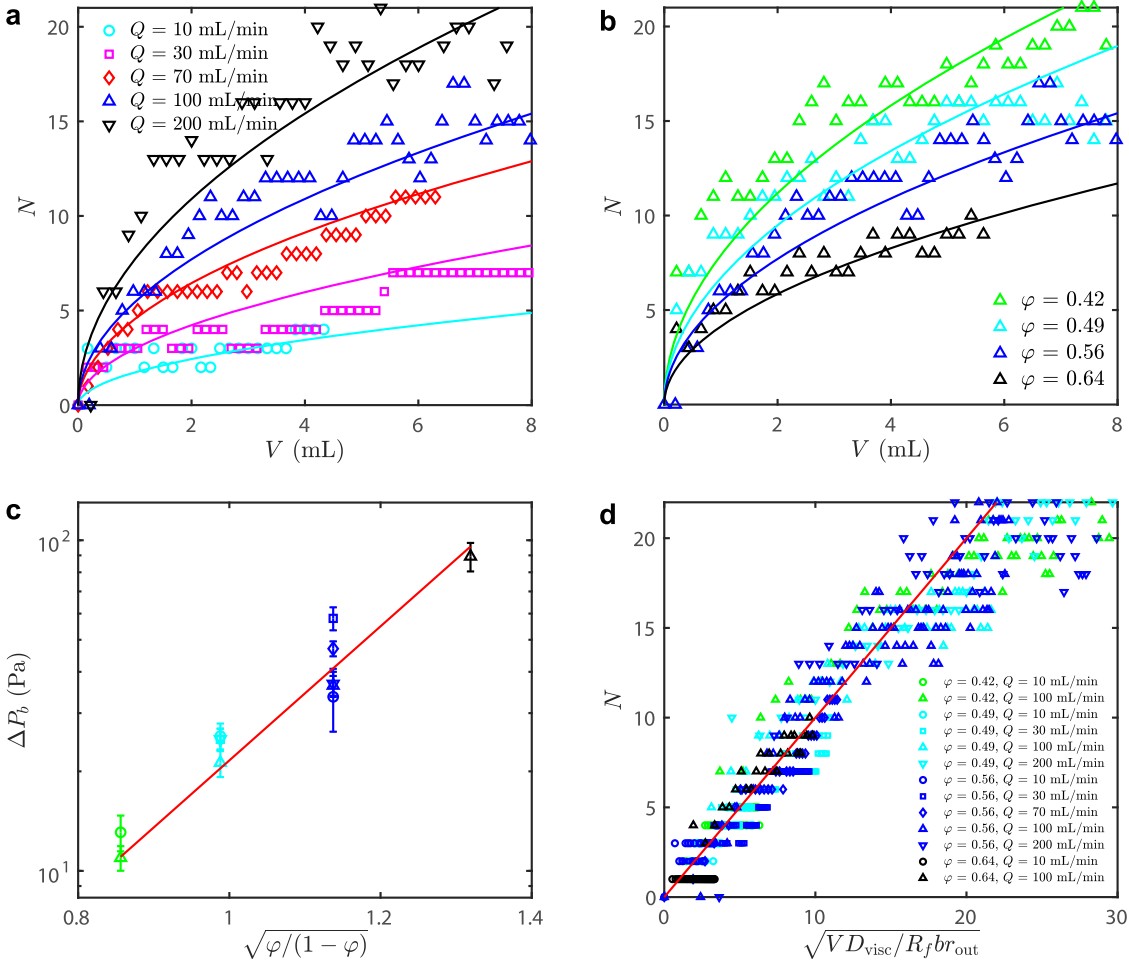

**Fig. 5 | Simultaneously growing fingers.** $N$ as a function of injected volume $V = Qt$ for (**a**) $\varphi = 0.56$ and different injection rates $Q$ and (**b**) $Q = 100$ mL/min and different filling fractions $\varphi$. We also plot Eq. (7) for each $Q$–$\varphi$ combination (curves), fitting the value of $\Delta P_b(\varphi)$ in each case. **c** We then plot these values of $\Delta P_b(\varphi)$ against $\sqrt{\varphi/(1-\varphi)}$ on a semi-log scale (same legend as (**d**)), indicating an exponential relationship with $\sigma_\beta = 0.20$ (red line, Eq. (2)). **d** Finally, we plot $N$ against $\sqrt{VD_{\text{visc}}/(R_f b r_{\text{out}})}$ for all experiments, as suggested by Eq. (7) (red line).

fingers as $\varphi$ increases. This observation suggests that the number of fingers $N$ should decrease with $\varphi$, as is confirmed in Fig. 5b. To capture this feature, we follow previous work[15,39,40] by hypothesising an exponential friction law of the form

$$\Delta P_b[L_f(\varphi)] = \sigma_\beta \exp\left[\frac{L_f(\varphi)}{\xi}\right] = \sigma_\beta \exp\left[\frac{2}{\xi}\sqrt{\frac{\gamma b \varphi}{\sigma_0(1-\varphi)}}\right], \quad (2)$$

where we use the pre-factor $\sigma_\beta$ as a fitting parameter.

The viscous contribution to the liquid pressure vanishes at a moving finger tip and increases linearly with distance upstream of the tip. The viscous pressure $\Delta P_v$ a distance $\Delta r_b$ upstream of a moving finger tip can be estimated via Darcy's law,

$$\frac{Q}{2R_f b N} \approx \frac{b^2}{12\eta_{\text{inv}}}\frac{\Delta P_v}{\Delta r_b} \quad \rightarrow \quad \Delta P_v \approx \frac{6\eta_{\text{inv}}Q}{R_f b^3}\frac{\Delta r_b}{N}, \quad (3)$$

where $Q/(2R_f b N)$ is the average flux within each active finger. Thus, a new finger will form a distance $\Delta r_b$ behind a moving finger tip when $\Delta P_v \sim \Delta P_b$, suggesting that

$$\frac{\Delta r_b}{r_{\text{out}}} \sim \frac{N}{D_{\text{visc}}}, \quad (4)$$

where $r_{\text{out}}$ is the radial system size (i.e. the outer radius) and

$$D_{\text{visc}} = \frac{\Delta P_v(\Delta r_b = r_{\text{out}}, N = 1)}{\Delta P_b} = \frac{6\eta_{\text{inv}}Q r_{\text{out}}}{R_f b^3 \Delta P_b} \quad (5)$$

is the dimensionless viscous deformability, which compares the characteristic viscous pressure drop to the characteristic frictional resistance of the side walls for a single finger of length $r_{\text{out}}$. Note that $\Delta r_b$ decreases with $D_{\text{visc}}$, so that faster injection or a more viscous invading phase will promote branching closer to the tip.

To sprout a single new finger from an existing one requires that a volume $2R_f b \Delta r_b$ of invading fluid be added to that finger. Thus, the addition of a volume $\Delta V$ to the flow cell will sprout $\Delta N$ new fingers, where

$$\Delta N = \frac{\Delta V}{2R_f b \Delta r_b} \quad (6)$$

For large $N$, we approximate the discrete variation $\Delta N$ as a continuous one and integrate from $V = 0$, giving

$$N(V) = \sqrt{\frac{VD_{\text{visc}}}{R_f b r_{\text{out}}}}, \quad (7)$$

where we have eliminated $\Delta r_b$ using Eq. (4).

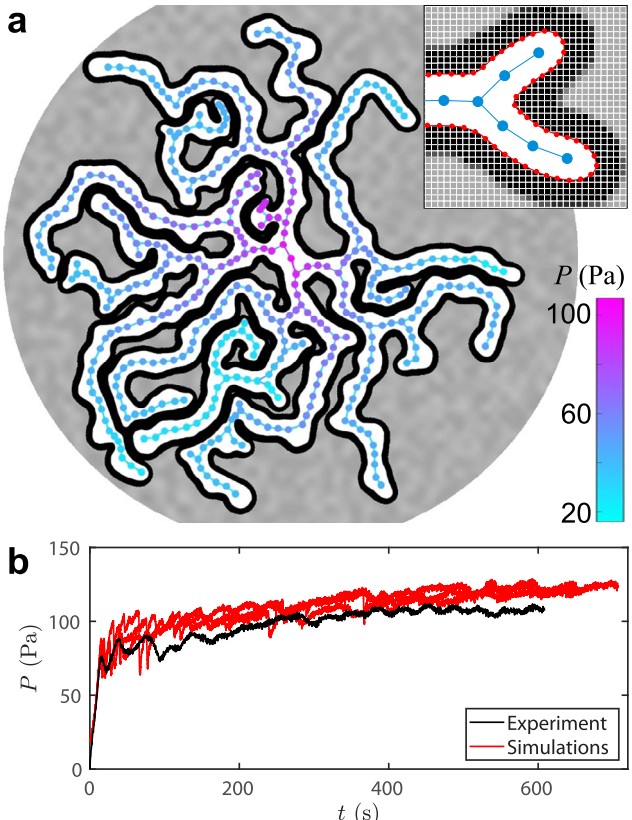

**Fig. 6 | Simulation of frictional fingers. a** $D_{\text{visc}} = 22$, corresponding to injection of water at $Q = 100$ mL/min with $\varphi = 0.42$. The viscous pressure field (cyan to magenta) is calculated on a branching tree that follows the skeleton of the fingers. The outer radius is $r_{\text{out}} = 13.4$ cm and the gap thickness is $b = 0.9$ mm. Inset: Close-up showing the filling-fraction field (grey-scale pixels), the interface (red chain of nodes), and the viscous skeleton (blue nodes and edges). **b** Fluid pressure at inlet for experiment and simulation at $D_{\text{visc}} = 31$ ($Q = 1$ mL/min, $\eta = 141.4$ Pa.s).

On the right-hand side of Eq. (7), the only unknown is $\Delta P_b$, which appears within $D_{\text{visc}}$ (recall that $R_f(\varphi)$ is given in Eq. (1)). We therefore plot the number of active fingers $N$ against the total injected volume $V$ from experiments at fixed $\varphi$ for several different values of $Q$ (Fig. 5a) and at fixed $Q$ for several different values of $\varphi$ (Fig. 5b). Then, we use $\Delta P_b(\varphi)$ as a fitting parameter to achieve the best match between these results and the predictions of Eq. (7). In Fig. 5c, we then plot these best-fit values of $\Delta P_b$ against $\sqrt{\varphi/(1-\varphi)} \propto L_f$, as suggested by Eq. (2); the linear trend of the data on this semi-logarithmic scale provides support for the exponential friction model for the side-walls (Eq. (2)). A least-squares fit of these empirical $\Delta P_b$ values against $\sqrt{\varphi/(1-\varphi)}$ (red line in Fig. 5c) suggests that $\sigma_\beta = 0.20 \pm 0.02$ Pa, where the 95% confidence interval of the fit is used for the uncertainty. Finally, we plot $N(V)$ against $\sqrt{VD_{\text{visc}}/(R_f b r_{\text{out}})}$ for all of our experimental results in Fig. 5d (symbols), along with the prediction of Eq. (7) that these two quantities are directly proportional (red line).

The simple model captures the overall trend in the data, despite containing none of the geometrical complexity of the branching patterns. Note, however, that there is considerable variability in the data. The actual strength of viscous forces within a finger depends on the finger width, which exhibits 10–20% variability around the characteristic value $R_f$ at the same value of $\varphi$ (Fig. 3). On the vertical axis, we observe that $N$ can vary both between different experiments and also within a single experiment as fingers start, stop, and sometimes restart again, making this measurement inherently imprecise. Note also that, for large $N$, the fingers begin to fill the available space, crowding out the formation of new fingers. This effect is not included in the model,

but should suppress the growth of $N$ at even higher values of $\sqrt{VD_{\text{visc}}/(R_f b r_{\text{out}})}$. Nevertheless, our simple model captures reasonably well the roles of viscosity and friction in controlling the sprouting of new fingers.

## Simulations of viscously stable frictional fingering

To better understand the complex patterns generated in experiments, we develop a frictional-fingering simulator by building on a code previously used to model air invasion into a wet hydrophilic packing[42,43]. The granular filling-level field is represented as a 2D array (grey-scale pixels in Fig. 6a). The interface between the invading fluid and the defending mixture is represented by a chain of nodes (red points in the inset of Fig. 6a). As in the experiments, forward motion of the interface accumulates a compaction front. To implement viscous flow inside the invading phase, we skeletonise the finger structure into a branching tree (blue nodes and edges in the inset of Fig. 6), on which we calculate the local viscous pressure $P_\nu$ (Eq. (12)). Flow through the viscous skeleton is determined by the volume change dictated by growth at the tips of the fingers. An interface segment advances incrementally when its yield pressure is exceeded by the local fluid pressure, after which the filling fraction, the local interface curvatures $R_{\text{local}}$, and the viscous pressures $P_\nu$ along the flow network are updated. Note that we neglect the viscosity of the defending fluid, and that we use two fixed fitting parameters, the friction coefficient and a viscous coefficient, both set to match finger widths and viscous stabilisation across the range of experimental parameters (see Methods).

Figure 6a shows a detailed view of a simulation at moderate $D_{\text{visc}} = 22$ with multiple active fingers. The skeletonised viscous flow network shows the fluid pressure, which decreases toward the tips of active fingers and is uniform along inactive fingers, with the maximum pressure at the inlet. Figure 6b shows the evolution of injection pressure for $D_{\text{visc}} = 31$, comparing the experimental pressure measured at the inlet with the simulation pressure calculated at the central node for three realisations. In both experiment and simulations, the pressure initially increases rapidly as bulldozing mobilises friction and capillary forces, then more slowly due to build-up of viscous pressure as fingers grow in length. The simulations reproduce the trend in the pressure data from experiments, as well as the typical frequency and magnitude of the fluctuations; however, the overall magnitude of the injection pressure is somewhat higher in the simulations than in the experiment. Figure 2b shows simulation results across a range of filling fractions and injection rates, reproducing the experimental transition from single-finger to multi-finger growth as a function of $Q$, although somewhat under-predicting the number of fingers at high $Q$. Figure 3 includes the simulated finger width $2R$ at low $Q$, which decreases a bit more steeply with $\varphi$ than what is observed in the experiments.

These differences between experiment and simulation may be due to the fact that the simulations use an exponential friction model (Eq. (11)) along the entire the compaction front, regardless of curvature. Our experimental data indicates that friction along the finger side walls is indeed well captured with an exponential model (Fig. 5), but that friction at the finger tips is better captured by a linear friction model (Fig. 3). We have used the exponential model in the simulations to prioritise viscous stabilisation, which is controlled by side-wall friction. However, tip friction controls the finger width and the resistance to finger propagation, so it is not surprising that the simulations exhibit a steeper variation of $2R$ with $\varphi$ and a larger injection pressure than observed in experiments. Resolving this discrepancy would require the use of curvature-dependent friction along the compaction front, which may be the subject of future work.

## From individual fingers to radial spoke patterns

To further increase the strength of viscous stabilisation relative to capillarity and friction, we increase the viscosity of the invading

fluid. Figure 7 shows the time evolution of glycerol injection at $Q = 10$ mL/min, producing extreme viscous stabilisation and fingers that radiate outward in an axisymmetric spoke pattern. Here, the frictional instability produces fingers and viscous stabilisation forces

them to grow radially, with an axisymmetric viscous pressure field. The tips remain equidistant from the inlet, creating a circular displacement front with embedded radial streaks of packed grains. As the pattern expands over time, the fingers increase in number by splitting to populate the growing circumference while maintaining a constant characteristic finger width (see Supplementary Movie 3).

Figure 8a shows an experimental $\eta_{inv}$–$Q$ phase diagram in a log-log plot. The viscosity of the invading fluid $\eta_{inv}$ (i.e. the fraction of glycerol) increases from bottom to top and the injection rate $Q$ increases from left to right. The top-right corner is empty because of the force limitation on the pump. Figure 8b shows a corresponding $\eta_{inv}$–$Q$ phase diagram for the simulations over a wider range of $Q$.

In these phase diagrams, similar patterns fall along diagonal lines corresponding to constant values of the product $\eta_{inv}Q$, which is a measure of the strength of viscous forces. We quantify the balance of viscous forces, which drive the motion of the grains, to friction, which resists the motion of the grains, via the dimensionless viscous deformability $D_{visc}$ introduced above (Eq. (5)), which captures precisely this balance.

Our $D_{visc}$ is similar in spirit to the large-capillary-number limit of the 'fracturing number' of Holtzman et al.[21], where the motion of a granular solid is resisted by friction under confining stress; to the 'viscous fracturing number' of Carrillo and Bourg[44], where the motion of a porous viscoplastic solid is resisted by a yield stress; and to the 'fluidisation number' of Campbell et al.[25], where the motion of a granular material was resisted by friction due to the weight of the grains. In the present context, friction is instead controlled by bulldozing, pile-up, and bridging.

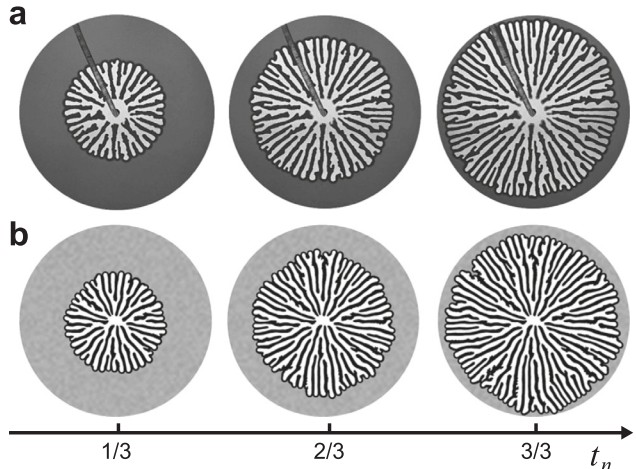

**Fig. 7 | Time evolution of spoke pattern. a** Experiment and (**b**) simulation of glycerol injection producing a viscously stable spoke pattern. Time $t_n$ is normalised by the time the first finger reaches the boundary. These results are for $Q = 10$ mL/min, $\eta_{inv} = 1414$ mPa·s, $\varphi = 0.49$, $b = 0.9$ mm, and $r_{out} = 13.4$ cm (see Supplementary Movie 3).

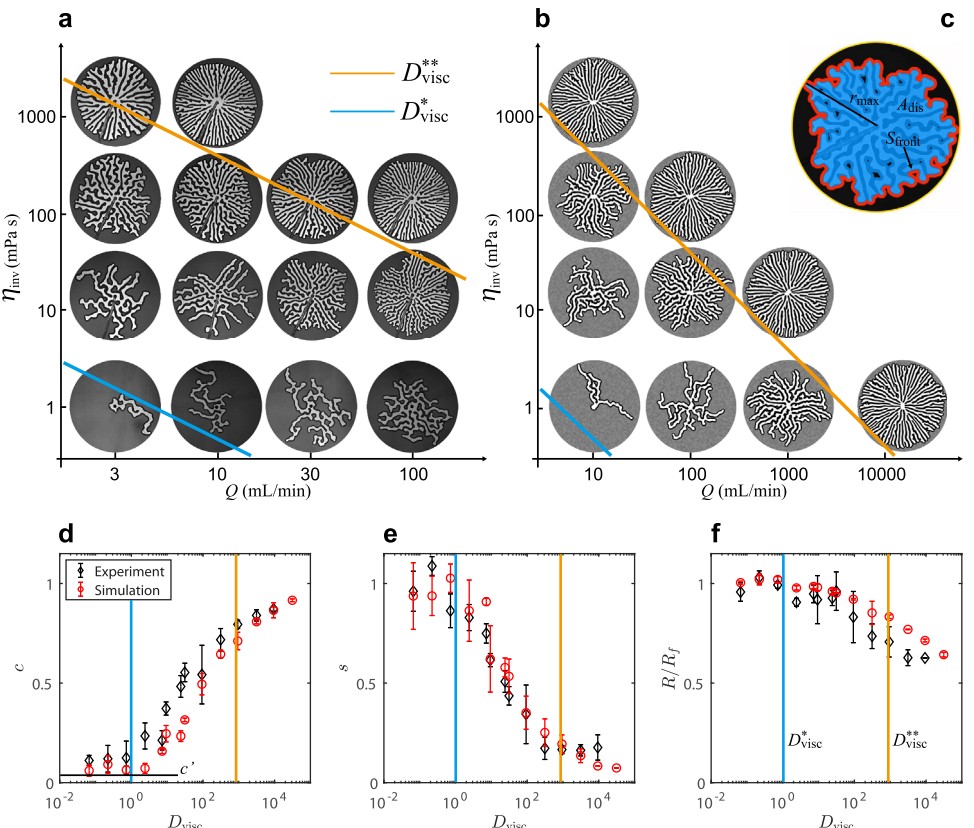

**Fig. 8 | $\eta_{inv}$–$Q$ phase diagrams. a** Experiments and (**b**) simulations of the invasion of water-glycerol mixtures into dry hydrophobic grains. The viscous deformability at which the pattern transitions from single to multiple fingers ($D^*_{visc}$) and from multiple fingers to radial spokes ($D^{**}_{visc}$) are plotted in blue and orange, respectively (Eqs. (8) and (9)). **c** Definitions of front length $S_{front}$ (red curve) and displaced area $A_{dis}$ (invaded area plus compaction front, blue region), with $r_{max}$ the reach of the most advanced finger. **d** Pattern compactness $c = A_{dis}/A_{reach}$, (**e**) front instability number $s = S_{front}/S_{finger}$, and (**f**) finger width $2R$ normalised by the characteristic rate-independent finger width $2R_f$ (Eq. (1)) as functions of $D_{visc}$. $D^*_{visc}$ and $D^{**}_{visc}$ in blue and orange lines, and theoretical lower $c'$ indicated in (**d**). In all panels, $\varphi = 0.49$, $b = 0.9$ mm, and the outer radius is $r_{out} = 13.4$ cm.

Viscous stabilisation relates to the two macroscopic length scales: the system size, represented by the cell radius $r_{out}$, and the emergent finger width $2R_f$. At low $D_{visc}$, a single finger may grow to reach the perimeter of the cell without spawning a new finger if $\Delta r_b$ is greater than the system size $r_{out}$. Taking $\Delta r_b = r_{out}$ and $N = 1$, Eq. (4) suggests that the transition between a single finger and multiple fingers should occur around

$$D_{visc}^{*} = 1, \tag{8}$$

which is ultimately the scenario that motivated our definition of $D_{visc}$. Note that $D_{visc}$ is proportional to system size, so a larger system requires a smaller viscosity or a lower injection rate to avoid branching.

The next transition, from multiple fingers to radial spokes, occurs when the critical distance $\Delta r_b$ becomes smaller than the characteristic finger width $2R_f$. Branching will then be suppressed by the presence of neighbouring fingers as the pattern becomes space-filling. These patterns are viscously stabilised in the sense that a finger that advances ahead of its neighbours will decelerate due to its larger internal pressure drop, leading to a spoke pattern where all fingers extend roughly the same distance $r$ from the inlet. The number of fingers $N_{spoke}(r)$ in this limit is easily estimated from conservation of mass, $N_{spoke}(r) = \pi r(1 - \varphi)/R_f$. Taking $\Delta r_b = 2R_f$ and $N = N_{spoke}(r_{out})$, Eq. (4) suggests that the transition to spokes should occur around

$$D_{visc}^{**} = \frac{\pi}{2}(1 - \varphi)\left(\frac{r_{out}}{R_f}\right)^2. \tag{9}$$

We indicate these transitional values in the $\eta - Q$ phase diagrams (diagonal blue and orange lines, respectively in Fig. 8a and b) for fixed $\varphi = 0.49$, $b = 0.9$ mm, and $r_{out} = 13.4$ cm, for which $D_{visc}^{**} = 860$. In both cases, we obtain a reasonable match to the visual characteristics of the patterns, transitioning from one finger to several around $D_{visc}^{*}$ and from multiple fingers to space-filling radial spokes around $D_{visc}^{**}$, although both transitions are clearly gradual. Note that the $D_{visc}^{*}$ and $D_{visc}^{**}$ lines (blue and orange, respectively) appear steeper in Fig. 8b than in Fig. 8a because the range of $Q$ is wider in the former.

For a quantitative analysis of the spatial characteristics of these patterns, we define the 'pattern compactness' $c = A_{dis}/(\pi r_{max}^2)$, where the displaced area $A_{dis}$ includes both fingers and compaction fronts (but not undisturbed material) and $r_{max}$ is the radial extent of the most advanced finger (Fig. 8c). For the measurements presented here, $r_{max} = r_{out}$. We plot $c$ against $D_{visc}$ in Fig. 8d, indicating the relevant values of $D_{visc}^{*}$ and $D_{visc}^{**}$ by the blue and orange lines, respectively. The theoretical lower limit $c' = 2R_f/(\pi r_{out}(1 - \varphi)) \approx 0.04$ corresponds to a single straight finger growing from the inlet to the edge. The compactness $c$ increases as viscous stabilisation creates multiple fingers and a more compact pattern, approaching 1 for radial spoke patterns.

We define a 'front instability number' $s = S_{front}/S_{finger}$, where $S_{front}$ is the length of the outer edge of the compaction front (i.e. the outer boundary between the compaction front and the undisturbed material; see red contour in Fig. 8c). For the spoke pattern, this boundary becomes nearly circular since the compaction fronts of neighbouring fingers touch. $S_{finger}$ is the longer internal perimeter of the finger pattern, tracing the liquid–air interface. The value of $s$ is close to 1 for a single finger where the outer front perimeter follows the internal finger interface, decreasing as fingers increasingly meet to form a common front (Fig. 8e).

The finger width $2R$ (Fig. 8f) is approximately $2R_f$ in the rate-independent regime (Eq. (1) and Fig. 3). Naively, one would expect increased viscous pressure within the finger to expand the width; instead, the fingers are observed to narrow when approaching $D_{visc}^{**}$. This narrowing is most likely a result of self-confinement, in which the competition between numerous fingers increasingly suppresses lateral

expansion. There is a gradual transition from multiple individual fingers to side-by-side radial spokes as viscous stabilisation becomes stronger and stronger, as evidenced by $2R$ beginning to decrease before the system reaches $D_{visc}^{**}$. Note that this self-confinement effect is not included in the model.

## Discussion

We have studied the fluid dynamics of an invading fluid displacing a defending fluid containing a sedimented granular material that is wetted by the defending fluid and repelled by the invading meniscus (drainage). For loose packings of grains, as in this study, capillary forces dominate over the weak frictional strength $G$ of the initial granular layer, $G = \mu_0 \rho_b g b \varphi$, where $\mu_0$ is the friction coefficient between the initial layer of grains and the plate, $g$ is the body force per unit mass due to gravity, and $\rho_b = (\rho_g - \rho_{def})(1 - n)$ is the bulk density difference between the granular layer and the defending fluid, where $n$ is the porosity of the packing. In other words, the "capillary deformability" $D_{cap} = (\gamma/d)/G$ of the system is large in all experiments presented here, $D_{cap} \gg 1$, such that the meniscus can easily bulldoze the grains into compaction fronts that are then frictionally unstable, creating frictional fingers.

Holding capillarity constant, we then explored the effect of viscous stabilisation using mixtures of water and glycerol as the invading fluid and air as the defending fluid (negative $\mathcal{M}$). The log viscosity contrast is large in all of our experiments ($1.7 < |\mathcal{M}| < 4.9$), such that pressure gradients in the low-viscosity defending fluid (the air) are negligible.

For low $Q$ and $\eta_{inv}$, viscous forces are negligible on the scale of the finger width. Increasing the grain filling level $\varphi$ increases the bulldozing friction and leads to narrower fingers. The finger width is set at the forward moving tip where a linear friction model $\sigma(L_f)$ produces a good fit to the data. Breakout of new fingers from the static side-walls is suppressed by the frictional resistance of the granular compaction front, which increases exponentially as a function of its thickness.

Increasing $Q$ or $\eta_{inv}$ increases the strength of viscous forces relative to frictional resistance, increasing the viscous deformability $D_{visc}$. Viscously stable displacement involves pressure gradients along the invading fingers, with pressure decreasing from the central inlet toward the finger tips. Viscous stabilisation manifests as the sprouting of new fingers once the frictional 'breakout' pressure $\Delta P_b$ of the walls is exceeded.

Two mechanisms determine the role of viscous stabilisation: (1) the strength of viscous pressure drop relative to frictional stress, as measured by $D_{visc}$, and (2) the distance between the central inlet and the finger tips. We have identified two critical threshold values of $D_{visc}$ that separate different types of fingering patterns within the cell. Starting with a single finger at low $D_{visc}$, increasing $D_{visc}$ eventually leads to the first threshold value at which the viscous pressure along the finger grows large enough to cause branching before the finger reaches the outer boundary; this value depends on the size of the system, since a longer finger implies a larger viscous pressure drop for the same value of $D_{visc}$. As a result, larger cells would produce multiple fingers at lower $D_{visc}$. Further increasing $D_{visc}$ leads to the second threshold value $D_{visc}^{**}$, at which the viscous pressure gradient within the fingers is large enough to produce breakout pressures immediately behind the finger tips. Fingers that move ahead of the pack are suppressed by their internal viscous pressure drop and new fingers sprout continuously to populate an ever increasing pattern perimeter (in a radial cell). Ultimately, the fingers grow side-by-side in a space-filling radial spoke pattern.

Figure 9 summarises the interplay between viscous stabilisation and friction in a $D_{visc} - \varphi$ phase diagram. Moving up the $D_{visc}$ axis increases number of active fingers and the compactness of the final pattern. Moving along the $\varphi$ axis increases the frictional resistance to breakout, suppressing the viscous stabilisation mechanism. The

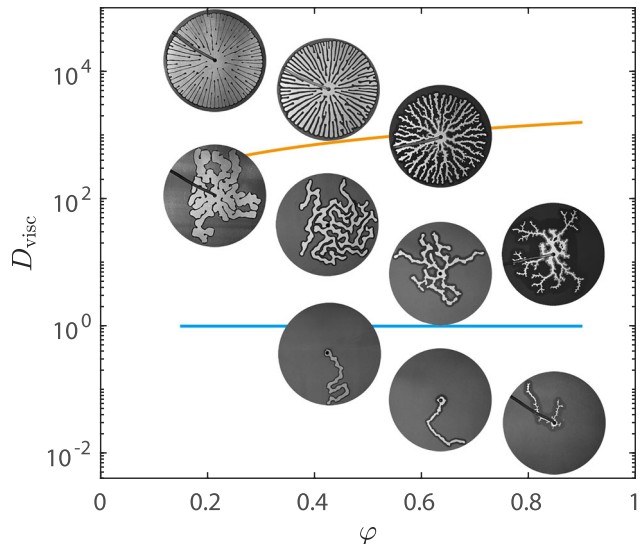

**Fig. 9 | $D_{\mathrm{visc}} - \varphi$ phase diagram.** Increased viscous forces produces increasingly compact patterns, while increasing $\varphi$ leads to thicker compaction fronts that suppress branching of new fingers, counteracting the stabilisation effect. Blue and orange lines represent $D_{\mathrm{visc}}^{*}$ and $D_{\mathrm{visc}}^{**}$ transitions from single to multiple fingers, and to spokes.

estimated transitions from single to multiple fingers $D_{\mathrm{visc}}^{*}$ and from multiple fingers to spoke pattern $D_{\mathrm{visc}}^{**}$ are plotted in blue and orange, respectively. Both transitions are gradual and the system exhibits a fairly large degree of variability, so neither value of $D_{\mathrm{visc}}$ represents a sharp phase boundary; rather, they are indicative of the expected location of the transition region. As such, these transitional values agree reasonably well with the evolution of the geometric features (Fig. 8d–f) and visual characteristics (Fig. 9) of the patterns.

Finally, it is instructive to consider our results in the context of traditional fluid displacement problems. As noted in the introduction, the corresponding fluid–fluid problem in a rigid Hele-Shaw cell is viscously stable and generates a circular displacement front in all cases. The defending phase in our system is a mixture of air and solid grains, so one might naively rationalise this frictional-fingering instability by thinking of the mixture of air and solid grains as a complex but effectively highly "viscous" defending phase, in which case the system would indeed be susceptible to a viscous-fingering-like instability. However, this naive view is directly contradicted by the fact that our system is unstable at low rates and increasingly stabilised by viscosity at higher rates. In a rigid porous medium, this problem (viscously stable drainage) does have a pattern-forming mechanism at low rates that is stabilised by viscosity at higher rates, but those invasion-percolation patterns are fractal and controlled by pore-scale disorder; they are independent of any macroscopic physical properties and, by definition, lack a characteristic macroscopic length scale. In contrast, our problem features an emergent macroscopic finger-width that is much larger than the grain size and that varies in a predictable way with macroscopic interfacial tension, gap thickness, filling fraction, and frictional resistance (Eq. (1)). The fingering patterns themselves are weakly influenced by random spatial variations in the initial filling fraction, but insensitive to variations at the grain/pore scale.

Multiphase frictional flows are thus a distinct class of fluid displacement problems in which pattern formation is controlled by capillarity, viscosity, and both inter-granular and sliding friction. Drainage at large capillary deformability and strong mobility ratio is now relatively well understood for both stable and unstable scenarios (e.g. Ref. 19 and the present study, respectively), but much of the parameter space remains unexplored.

## Methods
### Experiments
The Hele-Shaw cell comprised two $40 \times 40 \times 1.5$ cm glass plates separated by a gap thickness $b = 0.9$ mm. A 6 mm diameter hole through the centre of the top plate provided an inlet. The invading fluid was injected at controlled volume flow rates $Q$ between 0.3 and 200 mL/min using a syringe pump (Harvard Scientific, PHD Ultra). The cell was back-lit, and images were recorded using a Nikon 1 J2 digital camera at 30 fps.

The internal surfaces of the cell and the granular material were rendered hydrophobic by a silanization procedure following[45]. The silanization solution was a mixture of Trimethoxy(octadecyl)silane (OTMS) and Isopropyl alcohol(IPA). The silanization process was as follows: (1) the OTMS and IPA was mixed together in the ratio of 1:100; (2) the pH of the solution was adjusted to 3 by adding diluted Sulfuric acid ($H_2SO_4$, 0.1 M) to promote the hydrolysis of OTMS; (3) the solution was stirred using magnetic stirrer for at least 60 min at room temperature to form a alkylsilanol solution.

The glass pate surfaces were treated by pouring alkylsilanol solution on the surface and wiping over several times to make the coating uniform. The subsequently dried glass plates were hydrophobic with an estimated air/water contact angle of 120°. Silanization procedures were performed inside a fume hood.

The granular material was made hydrophobic by silanization treatment of soda-lime glass beads (Honite 18). The glass beads were acid-cleaned prior to silanization by the following steps: first, glass beads were immersed in hydrochloric acid (HCL, 0.1 M) and stirred using magnetic stirrer for at least 1 h. Then, they were rinsed thoroughly with deionized water and oven dried at 80 °C. The dried beads were then sieved to a diameter range of 75–100 $\mu$m. The sieved beads were immersed into the silanization solution in a beaker and heated on a hotplate to accelerate the evaporation of the solution. The coated hydrophobic beads were sieved again to ensure no beads were clumped together.

Preparation of the granular layer: The dry hydrophobic beads were spread out on one of the treated glass plates (later to form the bottom surface of the Hele-Shaw cell). In order to achieve a granular layer of uniform thickness, two strips of adhesive tape were placed along opposite sides of the bottom plate, and a straight-edged tool resting on both tape strips was used to scrape the granular material into a uniform layer along the plate. The top plate was then mounted on top, separated from the bottom plate with 0.9 mm spacers. We varied the packing height by changing the thickness of the tape strips. Each strip consisted of several layers of tape film attached on top of one another. The tape film thickness was 63.5 $\mu$m, and between 6 and 12 layers were used to create strips producing granular layer heights $h$ between 0.38 and 0.76 mm, corresponding to $\varphi$ between $0.42 \pm 0.01$ and $0.84 \pm 0.01$, with the values verified and uncertainty estimated from trials where layers were made and then the mass measured independently. Experiments with lower filling fraction ($\varphi = 0.21$) are shown for formation of spoke patterns (Fig. 9), but note that filling fractions below 0.42 were not included in quantitative analysis of finger widths because of practical problems achieving uniform layer thickness for the thinnest layers. The cell was clamped together firmly after assembly to prevent the top plate from lifting. All four edges of the cell were left open to the atmosphere.

Different viscosities were achieved by mixing glycerol and deionized water, with glycerol volume percentages of 0%, 58%, 84% and 100% corresponding to viscosities of 1, 14.14, 141.4 and 1414 mPa·s respectively[46].

### Simulations
The numerical code builds on the frictional fingering simulation presented in[39,43]. This code uses a two-dimensional array of values to represent the height of the sediment layer in the cell at each point in

space, and a chain of nodes to represent the interface of the invading fluid, as illustrated in Fig. 6. Every timestep the modified threshold pressure $P_t$ is calculated for every interface node, following

$$P_t = \frac{\gamma}{R_{\text{local}}} + \sigma + P_v \qquad (10)$$

where $R_{\text{local}}$ is the local interface radius of curvature, $\sigma$ is frictional stress resisting motion and $P_v$ is the viscous pressure difference between the inlet and that point. $\sigma$ is expressed as in[15]:

$$\sigma = \frac{g\rho_b b}{2\kappa}\left[(\kappa\mu + 1)\exp\left(\frac{2\mu\kappa L}{b}\right) - 1\right] \qquad (11)$$

with $L$ being the distance to the nearest point on the array which is not yet fully filled. The values used for bulk density was $\rho_b = 1450 \text{ kg/m}^3$, and $\kappa = 0.58$ is the Janssen's coefficient. A static friction coefficient $\mu_s = 0.91$ is used, with the dynamic friction coefficient $\mu_d = 0.9$ being substituted in the exponent if the interface node has moved in the previous 500 cycles.

The viscous pressure difference $P_v$ between the inlet and each point is calculated each timestep on a simplified version of the finger pattern, reduced to a branching tree of nodes as illustrated in Fig. 6. New nodes are added dynamically to this tree during the simulation whenever any section of interface is deemed to be too far away from its nearest node, to ensure that the shape of the tree closely mimics the shape of the invasion pattern. Each interface node reads $P_v$ from its nearest node on this tree. Each node calculates $P_v$ according to the Hagen-Poiseuille equation as

$$P_v = P_v' + \frac{C6\eta Q_f X}{Rb^3} \qquad (12)$$

where $P_v'$ is the $P_v$ of its parent node downstream, $\eta$ is the viscosity, $Q_f$ is the flow rate into the finger downstream (averaged over the last 250 timesteps), $X$ is the distance to its parent node and $R$ is the average half-width of the finger between itself and its parent node. We use a calibration factor $C = 1.33$ to match the transitions in the simulations to the experimental observations. To calculate $R$, a two-dimensional array holds a value $x$ for each position in the cell, $x$ being the distance to the nearest interface; this array is updated whenever the interface advances. $R$ is then estimated by stepping backwards towards the parent node along the ridge of the $x$ distribution, taking the mean $x$ along that ridge as the value of $R$.

Once the node with the lowest $P_t$ is identified, it is advanced forward slightly; its three nearest neighbours on each side are also moved by a lesser distance to maintain a smooth interface. New nodes are interpolated into the interface chain when the spacing between nodes exceeds a critical threshold, to maintain interface resolution as the interface lengthens. Whenever the interface advances, all granular material from the invaded region is redistributed to the nearest uninvaded positions which have space available. Randomness is introduced by initialising the distribution of granular material with random fluctuations above and below $\varphi$.

The 250 timestep period for averaging flow rate and the 500 timestep period for transitioning from kinetic to static friction are arbitrary numbers. They were chosen for being large compared to the typical number of growing fingers (to avoid excessive discretisation of viscous pressure and to prevent slower-moving but still active fingers from taking on static friction), but still small compared to the typical time period of an entire simulation. Qualitative tests did not suggest that the fingering patterns were strongly sensitive to these parameters. The simulated patterns match the experiments reasonably well across a wide range of $D_{\text{visc}}$ and $\varphi$.

## Data availability

Images from experiments and simulations are available on the Zenodo data repository with the https://doi.org/10.5281/zenodo.7890690[47].

## Code availability
The Python code for the simulations are available from the authors upon request.

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

## Acknowledgements

We thank Miles Morgan, Deren Ozturk, Rowan Brown and Duncan Hewitt for discussions. This research was funded by the Engineering and Physical Sciences Research Council [EP/S034587/1] (B.S., C.W.M.); the European Research Council (ERC) under the European Union's Horizon 2020 Programme [Grant No. 805469] (C.W.M.); the China Scholarship Council (CSC) (D.Z.); and the Research Council of Norway through its Centres of Excellence funding scheme [262644] (E.G.F., K.J.M.).

## Author contributions

D.Z. performed the experiments and analysed the data. J.M.C. and J.A.E. developed the numerical simulations. J.M.C. and D.Z. calibrated the code and performed the simulation parameter study. E.G.F., J.M.C., C.W.M. and B.S. developed the theory. E.G.F., K.J.M., C.W.M. and B.S. contributed to supervision, discussions and proof-reading. B.S. conceived the study, and D.Z., J.M.C., E.G.F., C.W.M. and B.S. contributed to writing the paper.

## Competing interests

The authors declare no competing interests.
