## [Peer Review File · Nature Communications]

REVIEWER COMMENTS

Reviewer #1 (Remarks to the Author):

The authors study viscously stable and viscously unstable frictional flow patterns in drainage using experiments and simulations. The paper is well written and the figures generally good. I had thought that Nature Communications placed a limit on figures well below the 9 figures displayed in the main text of this manuscript, but maybe I misunderstood or repackaging is necessary. More importantly, in the experiments in a narrow gap between two plates (Hele-Shaw configuration), air is displaced by a viscous fluid (water or a glycerol-water mixture) at a given flow rate with a given area fraction of beads (grains). The grains are free to reorganize, which I think is one of the distinguishing characteristics of this work. The authors highlight the role of friction (σ_0) though whether this is with other beads or beads on the plate is left unclear, and the role of granular compaction/friction through the parameter G that enters the parameter D . The main findings are apparently stated on p. 3: "When the flow is viscously stable, increasing D_{visc} leads to a striking transition from the growth of one solitary finger to the simultaneous growth of multiple, wandering fingers to the axisymmetric growth of a radial spoke pattern as the flow is increasingly viscously stabilised. When the flow is viscously unstable, in contrast, the invasion patterns transition from frictional fingering to classical viscous fingering as D_{visc} increases beyond a critical fluidisation threshold." Organizing the results via the dimensionless parameter D seems like a good advance, though to this reviewer's eye figure 7 does NOT provide a convincing demonstration that the parameter explains similarities and differences in the patterns. Maybe this paper merits further consideration in Nature Communications but first the authors have to clarify what features are really new, and whether these are details of a well-studied system or novel features, since they have failed to reference, or even acknowledge, what seems to this reviewer other related studies of what may be the main features of the experiments and simulations, several certainly showing experiments and simulations of very similar systems.

1. p. 2: We read "Without the grains, reversing the two viscosities ($\eta_{\text{def}} < \eta_{\text{inv}}$) negates the fingering instability by turning viscosity into a stabilising force. With the grains, however, the flow remains frictionally unstable due to bulldozing. The competition between these two mechanisms has not previously been studied in any detail". On one level this is surely wrong. Already 30 years ago Lenormand investigated very similar problems, summarizing a wide variety of results in plots of a viscosity (or mobility) ratio M versus the capillary number, e.g., Lenormand (1990), Liquids in Porous Media, J. Phys. Cond. Matter. There you already find experiments with mercury displacing air in a drainage configuration. A wide variety of results are given in Figure 9 in that paper which is a drainage phase diagram. This work has a large literature as Lenormand's papers have been cited hundreds of times (one of them has been cited >1400 times). None of Lenormand's work is cited but already that work showed some of the main features (though maybe for a fixed porous medium) as shown in the paper under review.

2. The work cited in item 1 has been extended in several directions in recent years, e.g., see the uncited paper by Primkulov, Pahlavan, Fu, Zhao, McMinn and Juanes, Wettability and Lenormand's diagram, *J. Fluid Mechanics*, 2021, where the authors extend Lenormand's diagram with the impact of wettability using dynamic and quasi-static pore-network models. None of these references are cited by the authors, so I simply disagree with the main claim of the paper that "The competition between these two mechanisms has not previously been studied in any detail" – it has been studied in a lot of detail though as with any topic there can be some avenues yet to explore. I guess the main difference with other work is that the grains can rearrange or varying the area fraction, but the authors have not done a good job of helping a non-expert understand what are the new contributions to the field. Even more significantly, papers such as this one and others by Juanes and colleagues (and the many references) show many experiments and simulations similar, at least qualitatively, to the pattern formation figures in the present paper.

3. There are further studies of again what seems like related ideas, again uncited. For example, Zhao et al. (Juanes group) in *PNAS* in 2019, "Comprehensive comparison of pore-scale models for multiphase flow in porous media." Again, there are many similar experiments and simulations compared to the work in the present paper, but it may be that the significant advance is better understanding/organization of the patterns, qualitatively or possibly quantitatively (?) through the parameter D but the authors do not help themselves by not making clear, as far as I can tell, the microstructural features that distinguish their viscous invasion flows from previous work.

4. Figure 7 – Doesn't this figure have many similarities (except that it is dimensional) with the dimensionless phase-like diagrams given by Lenormand (and many others), and more recently in the extensions discussed by Juanes and colleagues?

5. Haven't figures such as figures 6 and 9 appeared many times in papers by others in multiphase flow in porous media of different kinds? What is new to this paper under review? What new understanding is transmitted by the figure? This is the authors' responsibility; it should not be the reviewer who is trying to figure out what is the distinguishing feature of the different figures.

6. Figure 8 looks reasonable but it does NOT show that the authors theory is predictive since it only demonstrates results for one value of ϕ and one plate spacing b and even then the trends are not convincing, though qualitatively they are suggestive.

7. The friction stress σ is said to be a function of R but on dimensional and physical grounds shouldn't it also depend on speed? For example, figure 3 is unconvincing since only one flow rate is shown.

8. It is unclear to me what the authors mean by the “front width”. I see an indication in figure 1 but it is a little vague to me but maybe they never need to be precise (though they give a formula on page 4). With a little digging I understand that the idea and the formula given by the authors was explained already in their work (cited) from 2007-2008.

I have other possible questions but given my remarks above I think the paper surely needs a revision and better argument, at least for Nature Communications, about the relation of the work to other contributions in the field. So many of the figures in this work appear similar to other work in the field, some going back decades, I think the authors have to do better to highlight what parts of their work require the microstructural detailed understanding and then do better to show in what sense the microstructural understanding predicts the experimental results. With the latter, since the system is complex, I can appreciate that some of the understanding/prediction may be more qualitative than quantitative, but the current draft falls short in my view, or at least in my understanding at this time.

Reviewer #2 (Remarks to the Author):

The paper studies pattern formation in displacement flow between two narrow plates. Viscous fluid either invades space filled with hydrophobic granular particles, or is being seeded by hydrophilic particles and evacuated from this space by invading air. The problem is not too dissimilar from other systems involving front propagation, and explores interplay between physical mechanisms of pattern formation that combine the influence of frictional, viscous and surface tension forces. Ultimately, the system evolves to choose the path of least resistance, which of course is not too surprising, though some of the observed transitions between patterns are novel.

However, I do think that these transitions could and should be explored more thoroughly, and I am not convinced about some of the explanations in the paper (see below). Section III on viscously unstable patterns in particular is very superficial and limited in scope - three types of patterns are presented, but no attempt was made to quantify and therefore properly explain transitions between them, and the physics of the problem are certainly more complicated than the authors allude to in the discussion. A very limited (and presumably preliminary) experimental study was presented in this regime, so I don't see much point in including it in the current paper at all. Given that the paper overall needs more work at this stage, I do not think that it should be accepted for publication.

Regarding the viscously stable part:

I don't understand the arguments for setting finger width on page 5, which suggest that viscous pressure does not matter. Firstly, I don't think that a single finger case is special - as author indicate themselves later in the paper "branching will always occur if the system is large enough", the only question is how big is Δr_b . For any given pattern there seems to be a unique R set, so how come one can neglect viscous resistance. Are you saying that everything is set at the tip - but than the tip has different pressure depending on where the finger is located. Later on, on page 14, you state that "The finger width is independent of D_{visc} for individual fingers but" decreases with D_{visc} as self-confinement increases. But both are function of σ_0 , so how can that be? Also, why not do experiments at much smaller ϕ in figure 3 to confirm relationship (2): the range of ϕ over which the relationship is fitted is not convincing enough.

The relatively simple model used in numerical study might be ok, but I am really perplexed by the pressure scale obtained using it and reported in Figure 5 (and also a relatively low pressure used for fitting (7)). I find this really surprising having had experience with these system, so I would have expected to see some pressure measurements in experiments. I would strongly recommend that the authors do so, at least for some of the patterns. Simple back of the envelope calculations (using viscous fluid invading air) suggest that the pressure should be higher if I am not mistaken. The model also systematically under-predicts the number of fingers in Figure 4 (at least for higher Q), but that is not commented on or discussed anywhere.

Finally, I am not sure about (11). How is the finger length 13.4cm measured? Based on the instantaneous pattern that you have in figure 7? But then the nature of that pattern would not change if it was measured earlier in experiment (and a different r was therefore obtained). In that case, the threshold (11) is somewhat meaningless, and, as you have pointed out yourselves, more gradual.

Reviewer #3 (Remarks to the Author):

This paper looks at the competition of frictional and viscous forces when a fluid displaces a sedimented layer of particles in a confined geometry. In typical experiments, either the frictional or viscous limits of fingering are investigated, but this article investigates the transition from friction dominated to viscous dominated patterns when the invading fluid is both less and more viscous than the defending fluid. The result of these experiments is a suite of striking patterns that form, all accompanied by simulations that recapitulate the experimental results as well as theory that captures the essence of the physical effects. The text, while being a bit wordy at points, clearly lays out a physical intuition for the observed phenomena and shows how the transitions in pattern morphology occur when a viscous length scale becomes comparable to a frictional length and then

the system size. This study nicely bridges the displacement of dry granular particles to the classic experimental system of viscous fingering.

Overall this is a high-quality, interesting, and thorough study. I have a few questions and comments for the authors, detailed below. I do have a concern about several images in different figures that appear to have been manipulated in some way (see Major comment 3), though they do not seem to impact the results shown. However, I do recommend this for publication after minor revisions.

Major comments:

1. In Section II the authors derive an expression for the finger width that is independent of the viscous stresses in the low flow rate/ Ca number regime. Later in Sections II B and D, this same expression for the fingers is used when discussing patterns where new fingers emerge. When fingering occurs there is (as stated by the authors) a pressure build-up in the invading fluid that becomes comparable to the threshold pressure used to derive Eqn. 2. Please justify why, in the cases where it appears the viscous stress is of the same order as the frictional stress, the viscous stress can still be ignored when determining the finger size.

2. There is a striking difference in the interface of the fingering patterns in the +M regime and the -M regime, notably that the +M regime is able to exhibit fluidization of the grains and not form a compacted layer of grains. Naively, I would assume that as D_{visc} gets larger (particularly as G becomes small) that this could be achieved in the -M case as well. Do you ever expect there to be a regime where there could be uniform, circular displacement ($c=1, s=1$) of the grains in the case where they are hydrophobic? If the +M case could also fluidize the granular bed then it would be more clear that the transitions in the +/-M cases are similar. Also, in this light, I hope the authors could comment on the importance of hydrophobicity versus hydrophilicity in the grains and if this is crucial for the observed physics.

3. There are several instances where two panels in different figures are the same image, but are rotated mirror images of each other. The ones I noticed were: (i) Fig. 2a second row, the second image and Fig. 7c bottom row, second image, (ii) Fig. 6 a the top left image and Fig. 7a the lower left image, and (iii) four images of Fig. 7c and Fig. 9d the entire top row. I understand that you are showing the same experimental results in different contexts, but I am unsure why the images are manipulated in such a way, especially in showing a mirror image.

Minor comments:

1. There are a few places where it would be nice to have the numbers for parameters in the main text. In Section IIA when the experiment is introduced, please write the gap spacing. This is useful so that when the particle size and finger width are mentioned the reader has a sense of the relative size of the system. It also makes estimation of different pressures (P_b , P_t) more accessible for the reader to check.

2. At two points in the text you have fit parameters that could have some physical interpretation: σ_0 on page 5 and ΔP_b on page 7. Does the value of σ_0 make sense with friction of dry grains? The value of $\Delta P_b = 30$ Pa is very similar to P_t from Eqn. (1) if you calculate that quantity from the data in Fig. 3. If we interpret P_t as the pressure needed to move a compacted, static region of beads, then having that same pressure difference (ΔP_b) away from the tip of a finger cause new growth seems reasonable. Or is there a different interpretation for this?

3. On line 391 you mention that conservation of mass gives you another estimate of ϕ , but this expression assumes that the packing fraction of the undisturbed bed and the compacted region of grains are the same. The density of random granular packings can be sensitive to their method of preparation, is there evidence that the two regions have the same packing fraction?

4. For Eqn. 16 a fit value of $C=8$ is used, is there some justification for having this be a fit parameter instead of using $C=6$ derived from Poiseuille flow in a Hele-Shaw cell? Also, I'm concerned about the units of Eqn. 16, if C is dimensionless then X also needs to be so: is X normalized by the gap length?

5. The authors have included fantastic videos as supplementary material, it may be nice to actively reference them in the text so the readers are more aware of their existence.

We thank the reviewers for their detailed and constructive feedback, and for their further consideration. In the responses below, the comments of the reviewers are included verbatim in black and our replies follow in blue.

Reviewer #1 (Remarks to the Author):

The authors study viscously stable and viscously unstable frictional flow patterns in drainage using experiments and simulations. The paper is well written and the figures generally good. I had thought that Nature Communications placed a limit on figures well below the 9 figures displayed in the main text of this manuscript, but maybe I misunderstood or repackaging is necessary.

We thank the reviewer for their positive assessment. Note that the “Guide to authors” for *Nature Communications* states that an article may have up to 5000 words and up to 10 display items.

More importantly, in the experiments in a narrow gap between two plates (Hele-Shaw configuration), air is displaced by a viscous fluid (water or a glycerol-water mixture) at a given flow rate with a given area fraction of beads (grains). The grains are free to reorganize, which I think is one of the distinguishing characteristics of this work.

Indeed, the emerging flow patterns are a direct consequence of the feedback between the flow and local reorganisation of the frictional material. As discussed in more detail below, we believe that our revised Introduction and Discussion now do a better job of highlighting this point.

The authors highlight the role of friction (σ_0) though whether this is with other beads or beads on the plate is left unclear, and the role of granular compaction/friction through the parameter G that enters the parameter D .

We thank the reviewer for noting that our description of the role of friction was unclear. We have now expanded this description, including some discussion of the role of microstructure. See, for example, paragraphs 4 and 5 in Section II A:

“The grains in the compaction front bridge the gap between the plates. In a straight segment of the side-wall, the capillary pressure imparted by the meniscus is opposed by the effective stress in the granular material, which disperses through grain-grain contacts to the plates. Deformation of the front requires dilation, which is opposed by the confining plates and the granular pressure from neighboring front segments. Assuming Coulomb friction between the granular material and the plates and that out-of-plane stresses are proportional to the imposed in-plane stress (the “Janssen law” [40]), the frictional stress resisting motion of the front increases exponentially with front width: $\sigma(L) \propto e^{L/\xi}$, where $\xi = b/(2\mu\kappa)$ is the characteristic length, μ is the effective coefficient of

friction between the grains and the plates, and κ is the Janssen coefficient [39,41].

At the tip of a growing finger, the front is curved (Fig. 1c) and the streamlines of granular motion diverge as the grains are pushed outwards normal to the interface. In the reference frame of the moving tip, the granular material in the front is therefore continuously being stretched tangentially to the interface. This extensional motion reduces the granular pressure against the plates by weakening tangential granular force chains. The confinement-induced jamming that produced exponentially increasing friction at the straight side-walls is therefore significantly reduced at the moving tip, where the finger width is set. In agreement with previous work [42], we find that a simple linear friction model provides a good fit to the experimental data in Fig. 3 (see model results described below). Following [42], we assume a tip friction stress $\sigma_t = \sigma_0 L_t/b$, where L_t is the front width at the tip (see Fig. 1c) and σ_0 represents the friction stress per unit length b ; we use the latter as a fitting constant.”

The main findings are apparently stated on p. 3: “When the flow is viscously stable, increasing D_{visc} leads to a striking transition from the growth of one solitary finger to the simultaneous growth of multiple, wandering fingers to the axisymmetric growth of a radial spoke pattern as the flow is increasingly viscously stabilised. When the flow is viscously unstable, in contrast, the invasion patterns transition from frictional fingering to classical viscous fingering as D_{visc} increases beyond a critical fluidisation threshold.” Organizing the results via the dimensionless parameter D seems like a good advance, though to this reviewer's eye figure 7 does NOT provide a convincing demonstration that the parameter explains similarities and differences in the patterns.

We thank the reviewer for pointing out that the previous Figure 7 was unconvincing with regard to the controlling role of D_{visc} . We believe that this point was diluted in the original manuscript by the inclusion of the viscously *unstable* regime, in which the patterns are more complex and the role of viscosity more nuanced (but always destabilising). We have now removed the viscously *unstable* results from the manuscript in order to focus exclusively on the viscous *stable* results, which are the core novelty of our study. We have also revised the text and the figure (now figure 8) to provide a clearer and stronger case for the stabilising role of viscosity via $D_{\text{visc}} \sim \eta_{\text{inv}} Q$.

Maybe this paper merits further consideration in Nature Communications but first the authors have to clarify what features are really new, and whether these are details of a well-studied system or novel features, since they have failed to reference, or even acknowledge, what seems to this reviewer other related studies of what may be the main features of the experiments and simulations, several certainly showing experiments and simulations of very similar systems.

We thank the reviewer for highlighting the fact that we did not adequately frame our results in the context of previous work in the original manuscript. We previously framed our work in the context of fluid-fluid displacement in Hele-Shaw cells, which is a classical topic that has been studied extensively, and in the context of viscously unstable multiphase frictional flows, which attracted attention more recently. The reviewer is drawing attention here (and below) to the fact that we failed to compare and contrast with fluid-fluid displacement in *rigid porous media*, another classical topic in the literature on displacement patterns. We regret the omission and have amended it in the revised manuscript – particularly in the revised Introduction and the revised Discussion. As explained there and in response to the specific comments below, the work presented here is indeed entirely distinct. Viscously *stable* multiphase frictional flows have not been studied previously – They are a novel system with novel, striking features. Our manuscript presents the discovery of these flow patterns and a first theoretical framework to explain their characteristic features.

1. p. 2: We read “Without the grains, reversing the two viscosities ($\eta_{\text{def}} < \eta_{\text{inv}}$) negates the fingering instability by turning viscosity into a stabilising force. With the grains, however, the flow remains frictionally unstable due to bulldozing. The competition between these two mechanisms has not previously been studied in any detail”. On one level this is surely wrong. Already 30 years ago Lenormand investigated very similar problems, summarizing a wide variety of results in plots of a viscosity (or mobility) ratio M versus the capillary number, e.g., Lenormand (1990), *Liquids in Porous Media*, J. Phys. Cond. Matter. There you already find experiments with mercury displacing air in a drainage configuration. A wide variety of results are given in Figure 9 in that paper which is a drainage phase diagram. This work has a large literature as Lenormand’s papers have been cited hundreds of times (one of them has been cited >1400 times). None of Lenormand’s work is cited but already that work showed some of the main features (though maybe for a fixed porous medium) as shown in the paper under review.

We completely agree with the reviewer that viscously stable fluid-fluid displacements in rigid porous media have been studied extensively. We again thank the reviewer for pointing out that these flows make for a useful touchstone when framing our study. However, this observation does not contradict anything we wrote in the original manuscript, including the statement quoted by the reviewer and identified as “surely wrong”. In fact, it is **correct** that “the competition between these two mechanisms has not previously been studied in any detail”, where “these two mechanisms” are frictional instability and viscous stabilisation. Viscously stable fluid-fluid displacements in rigid porous media are controlled by the physics of capillary invasion, which are not active in our system at all because the invading fluid never penetrates the pore space; instead, capillarity bulldozes the grains to create open channels.

To clarify this confusion, we have broadened the framing of our work in both the revised Introduction and the revised Discussion. The second and third paragraphs of the Introduction now read:

“Multiphase frictional flows inhabit a large parameter space, but relatively few scenarios have attracted any attention. One such flow that is now relatively well understood is the injection of a low-viscosity invading fluid (viscosity η_{inv}) to displace a much more viscous defending fluid (viscosity $\eta_{\text{def}} \gg \eta_{\text{inv}}$) containing sedimented grains, for the case where the invading fluid is nonwetting to the grains (i.e., drainage). Without the grains, or with grains that are fixed in place (i.e., within a rigid porous medium), this problem is famously viscously unstable and will be subject to classical viscous fingering (i.e., the Saffman-Taylor instability) [32-35]. With movable grains, the nonwetting invading phase will tend to bulldoze the defending mixture rather than invading the space between or above the grains as long as capillary forces are strong enough to overcome friction with the wall(s) and among the grains (i.e., the capillary entry pressure must be sufficiently larger than the frictional resistance to sliding and rearrangement). This bulldozing behavior further destabilises the system as the accumulation of grains on the defending side of the interface penalises uniform displacement, leading to the formation of fractures, fingers, bubbles, labyrinths, and other patterns, depending on the injection rate and the packing fraction [15, 19, 28].

Without the grains, reversing the two viscosities ($\eta_{\text{def}} < \eta_{\text{inv}}$) negates the fingering instability by turning viscosity into a stabilising force. With grains that are fixed in place, capillary forces and pore-scale disorder compete with viscous stabilisation to produce fractal invasion-percolation patterns at low injection rates and rough but stable fronts at high injection rates [34-38]. With movable grains, however, the flow is frictionally unstable at all rates due to bulldozing. The competition between these two mechanisms has not previously been studied in any detail and even basic questions remain unanswered; for example, to what extent can viscous forces stabilise the flow against the frictional instability? Here, we explore this competition systematically using experiments and simulations. Focusing on the case where $\eta_{\text{inv}} \gg \eta_{\text{def}}$, we show that the pattern formation is controlled by the strength of viscous forces relative to friction, which can be quantified by a “viscous deformability” parameter D_{visc} . Increasing D_{visc} leads to a striking transition from the growth of one solitary finger to the simultaneous growth of multiple, wandering fingers to the axisymmetric growth of a radial spoke pattern as the flow is increasingly viscously stabilised.

In addition, the Discussion now includes the following paragraph:

“Finally, it is instructive to consider our results in the context of traditional fluid displacement problems. As noted in the introduction, the corresponding fluid-fluid

problem in a rigid Hele-Shaw cell is viscously stable and generates a circular displacement front in all cases. The defending phase in our system is a mixture of air and solid grains, so one might naively rationalise this frictional-fingering instability by thinking of the mixture of air and solid grains as a complex but effectively highly “viscous” defending phase, in which case the system would indeed be susceptible to a viscous-fingering-like instability. However, this naive view is directly contradicted by the fact that our system is unstable at low rates and increasingly stabilised by viscosity at higher rates. In a rigid porous medium, this problem (viscously stable drainage) does have a pattern-forming mechanism at low rates that is stabilised by viscosity at higher rates, but those invasion-percolation patterns are fractal and controlled by pore-scale disorder; they are independent of any macroscopic physical properties and, by definition, lack a characteristic macroscopic length scale. In contrast, our problem features an emergent macroscopic finger-width that is much larger than the grain size and that varies in a predictable way with macroscopic interfacial tension, gap thickness, filling fraction, and frictional resistance (Eq. 1). The fingering patterns themselves are weakly influenced by random spatial variations in the initial filling fraction, but insensitive to variations at the grain/pore scale.”

We believe that these changes will have thoroughly addressed the reviewer’s concerns with regard to novelty.

2. The work cited in item 1 has been extended in several directions in recent years, e.g., see the uncited paper by Primkulov, Pahlavan, Fu, Zhao, McMinn and Juanes, Wettability and Lenormand’s diagram, J. Fluid Mechanics, 2021, where the authors extend Lenormand’s diagram with the impact of wettability using dynamic and quasi-static pore-network models. None of these references are cited by the authors, so I simply disagree with the main claim of the paper that “The competition between these two mechanisms has not previously been studied in any detail” – it has been studied in a lot of detail though as with any topic there can be some avenues yet to explore. I guess the main difference with other work is that the grains can rearrange or varying the area fraction, but the authors have not done a good job of helping a non-expert understand what are the new contributions to the field. Even more significantly, papers such as this one and others by Juanes and colleagues (and the many references) show many experiments and simulations similar, at least qualitatively, to the pattern formation figures in the present paper.

With regard to the relationship of our results to previous work in rigid porous media, please refer to our previous responses. Once again, we completely agree with the reviewer that a comparison with fluid-fluid displacement in rigid porous media is useful for contextualising our work and results for a broad audience. The Introduction and Discussion now provide this context, citing several of the studies mentioned here, in the previous comment, and in the next comment. We are intimately familiar with these works — CWM is a co-author of several of them.

The main difference in our work here is indeed the fact that the grains can move, but this difference changes everything about the emergent patterns and the physics that control them. After reading the revised manuscript, we hope that the reviewer will understand and agree that the patterns presented here are only superficially similar to those presented in previous work. In fact, even the superficial similarity is limited.

3. There are further studies of again what seems like related ideas, again uncited. For example, Zhao et al. (Juanes group) in PNAS in 2019, “Comprehensive comparison of pore-scale models for multiphase flow in porous media.” Again, there are many similar experiments and simulations compared to the work in the present paper, but it may be that the significant advance is better understanding/organization of the patterns, qualitatively or possibly quantitatively (?) through the parameter D but the authors do not help themselves by not making clear, as far as I can tell, the microstructural features that distinguish their viscous invasion flows from previous work.

With regard to the relationship of our results to previous work in rigid porous media, please refer to our previous responses. However, we agree with the reviewer that the rationalisation of this system in terms of the controlling role of the parameter D_{visc} is indeed a key contribution of our work; in the revised manuscript, we have strengthened and clarified the latter aspects of our study.

4. Figure 7 – Doesn't this figure have many similarities (except that it is dimensional) with the dimensionless phase-like diagrams given by Lenormand (and many others), and more recently in the extensions discussed by Juanes and colleagues?

With regard to the relationship of our results to previous work in rigid porous media, please refer to our previous responses. Previous figure 7 (now figure 8) is a phase diagram for a pattern-forming process; the two axes are control parameters and the different data “points” show different patterns. The phase diagrams of Lenormand *et al.* 1988 and Primkulov *et al.* 2021, among countless others, share the same basic form – they are a standard tool for illustrating and rationalising pattern-forming processes. This is the full extent of the similarities between our diagram and those classical ones.

The axes of the classical diagram in Lenormand *et al.* 1988 for fluid-fluid displacement in a rigid porous medium are the dimensionless capillary number and mobility ratio; Primkulov *et al.* 2021 took the same diagram and added a third axis for contact angle. Both our patterns and our axes are different. Our two axes for this viscously stable multiphase frictional flow are the dimensional quantities η_{inv} and Q ; the point of the diagram is to show that the quantity $D_{\text{visc}} \sim \eta_{\text{inv}}Q$ is actually the relevant dimensionless control parameter. The only similarity between the two diagrams is that viscous stabilisation creates more compact patterns in both cases.

5. Haven't figures such as figures 6 and 9 appeared many times in papers by others in multiphase flow in porous media of different kinds? What is new to this paper under review? What new understanding is transmitted by the figure? This is the authors' responsibility; it should not be the reviewer who is trying to figure out what is the distinguishing feature of the different figures.

With regard to the relationship of our results to previous work in rigid porous media, please refer to our previous responses. We again thank the reviewer for highlighting the need to be explicit about these distinctions for the broad readership of a journal like *Nature Communications*.

6. Figure 8 looks reasonable but it does NOT show that the authors theory is predictive since it only demonstrates results for one value of phi and one plate spacing b and even then the trends are not convincing, though qualitatively they are suggestive.

This problem can be characterised by four *dimensionless* control parameters: the mobility ratio M , the capillary deformability D_{cap} , the viscous deformability D_{visc} , and the filling fraction ϕ . The study took the first two to be large, $M \gg 1$ and $D_{\text{cap}} \gg 1$, varied D_{visc} over a wide range, and performed a limited exploration of the impact of ϕ on finger width in the single-finger regime (Figure 3). However, we agree with the referee that ϕ deserved additional attention. For that purpose, we have re-analysed our data and performed some additional experiments to explore the impact of ϕ on the patterns, finding that a higher filling fraction produces thicker sidewalls, suppressing the breakout of new fingers and counteracting the mechanism of viscous stabilisation. We have also extended our basic theory to describe these new observations. The basic theoretical framework is now complete and in agreement with experimental results for a range of injection rate, viscosity and filling fraction. The results are presented in the revised Figures 3 and 5 and surrounding discussion.

Note that previous Figure 8 showed quantitative pattern characteristics to complement the qualitative visual patterns presented in previous Figure 7. The revised Figure 8 combines the patterns and the quantitative characteristics into a single figure.

7. The friction stress sigma is said to be a function of R but on dimensional and physical grounds shouldn't it also depend on speed? For example, figure 3 is unconvincing since only one flow rate is shown.

In fact, there is no dimensional or physical requirement that the frictional stress should depend on speed. Viscous resistance can be viewed as a rate-dependent friction, but dry granular friction is Coulomb-like and therefore rate-independent. A central result of previous studies on viscously unstable frictional-fingering is that the pattern characteristics and finger widths are rate-independent in the "frictional regime", below a critical flow rate. To demonstrate the applicability of this result here,

we have followed the reviewer's suggestion and included new data in the revised Figure 3 for other values of Q . The revised Figure 8f also shows additional measurements displaying constant R at low D_{visc} .

At higher and higher injection rates, we do eventually anticipate a transition to a more viscous-like frictional behavior due to fluidisation of the granular material (at some point, the grains begin to act like a very dense suspension). Fluidisation occurs when viscous stresses in the fluid around and between the grains start to become comparable to contact stresses. We are far from this threshold in our viscously stable experiments because the defending phase is air, for which viscous effects will only become important at very high rates. We do see some evidence of fluidisation at high rates in our viscously *unstable* experiments, but we have removed the viscously unstable results altogether for other reasons (see first response to Reviewer 2). This point suggests a promising avenue for future work.

8. It is unclear to me what the authors mean by the "front width". I see an indication in figure 1 but it is a little vague to me but maybe they never need to be precise (though they give a formula on page 4). With a little digging I understand that the idea and the formula given by the authors was explained already in their work (cited) from 2007-2008.

We thank the reviewer for raising this point. To clarify the definition of front width, we have added two more sub-panels to Figure 1 to more clearly illustrate the finger parameters and the viscous pressure gradient. We have also updated the caption accordingly, now stating that the front width L refers "to the part of the front that bridges the gap between the plates". It is difficult to obtain precise measurements for L because imaging parameters, lighting, and image-processing choices can influence the results. We do not attempt to present precise measurements of L and our models do not require this information.

I have other possible questions but given my remarks above I think the paper surely needs a revision and better argument, at least for Nature Communications, about the relation of the work to other contributions in the field. So many of the figures in this work appear similar to other work in the field, some going back decades, I think the authors have to do better to highlight what parts of their work require the microstructural detailed understanding and then do better to show in what sense the microstructural understanding predicts the experimental results. With the latter, since the system is complex, I can appreciate that some of the understanding/prediction may be more qualitative than quantitative, but the current draft falls short in my view, or at least in my understanding at this time.

We hope that our responses above and our changes to the manuscript will have thoroughly addressed the reviewer's concerns regarding the novelty of our work. We have also added a more detailed account of microstructural detail (see, e.g.,

paragraphs 4-6 in Section II A) and a Supplementary Information document including more detailed derivation and discussion of the models.

Reviewer #2 (Remarks to the Author):

The paper studies pattern formation in displacement flow between two narrow plates. Viscous fluid either invades space filled with hydrophobic granular particles, or is being seeded by hydrophilic particles and evacuated from this space by invading air. The problem is not too dissimilar from other systems involving front propagation, and explores interplay between physical mechanisms of pattern formation that combine the influence of frictional, viscous and surface tension forces. Ultimately, the system evolves to choose the path of least resistance, which of course is not too surprising, though some of the observed transitions between patterns are novel.

However, I do think that these transitions could and should be explored more thoroughly, and I am not convinced about some of the explanations in the paper (see below). Section III on viscously unstable patterns in particular is very superficial and limited in scope - three types of patterns are presented, but no attempt was made to quantify and therefore properly explain transitions between them, and the physics of the problem are certainly more complicated than the authors allude to in the discussion. A very limited (and presumably preliminary) experimental study was presented in this regime, so I don't see much point in including it in the current paper at all. Given that the paper overall needs more work at this stage, I do not think that it should be accepted for publication.

We thank the reviewer for the thoughtful comments. We included the viscously unstable flow regime in the original manuscript (previous Section III) for completeness, to give a broader picture of both stable and unstable flow configurations. However, the viscously unstable regime has been studied before in some detail, so a thorough presentation would necessarily require substantial repetition of previous results. The new experiments presented in the original manuscript were conducted in order to obtain results that are directly comparable with the new viscously stable results (radial cell, same granular material, same plate spacing, etc), but the description and analysis were minimised to avoid reiterating what is already known. We agree that this limited treatment left the impression that the work presented was not well balanced. To avoid distracting from the central theme of the present manuscript, the viscously *stable* regime, we have followed the reviewer's suggestion and removed the viscously *unstable* section altogether, including previous Section III and previous Figures 7c and 9. We have reworked the narrative to focus squarely on the viscously *stable* regime, including additional modelling, experimental results, and simulations to fully address the role of filling fraction. In the process, we believe that the manuscript has been greatly improved.

Regarding the viscously stable part:

I don't understand the arguments for setting finger width on page 5, which suggest that viscous pressure does not matter. Firstly, I don't think that a single finger case is special - as author indicate themselves later in the paper "branching will always occur if the system is large enough", the only question is how big is Δr_b . For any given pattern there seems to be a unique R set, so how come one can neglect viscous resistance. Are you saying that everything is set at the tip - but than the tip has different pressure depending on where the finger is located.

The reviewer raises an excellent point about the role of system size – if the system is large relative to Δr_b , then a new branch will always form eventually. We have clarified and emphasised this point in the revised manuscript, around Equation 8.

Separately, the viscous pressure along a finger is due to viscous flow through the finger. This viscous contribution is largest at the injection point and drops linearly with flow along the finger until it vanishes at the tip, where the pressure must exactly match the threshold for motion if the finger is moving. This threshold pressure is a combination of the capillary pressure from the meniscus and the friction from the bulldozed granular material (see text above Equation 1). Liquid invasion into a dry capillary tube without grains is exactly analogous – the viscous pressure must be maximum at the inlet and must vanish at the interface, where the pressure must balance the capillary pressure. We have revised Figure 1 to more clearly illustrate this point, including a new panel (d) that is specifically focused on pressure.

Because the threshold pressure is set by capillarity and granular friction, we expect it to be independent of both rate and position. (Both capillarity and granular friction become rate dependent at very high rates, the former because of dynamic-contact-angle effects and the latter because of granular fluidisation; all of our experiments are well below these transitions.) If multiple fingers of different lengths grow simultaneously, then all will share the same injection pressure and the same threshold pressure at the tip. As a result, a shorter finger must have a steeper pressure gradient along its length than a longer one.

We observe experimentally that the finger width is set at the tip, where the fluid pressure just balances the threshold pressure (see Figure 1b-d). The fingers do not widen upstream of the tip, even though the viscous pressure increases in that direction, because the straight, static side walls are harder to move than the curved, advancing tip (see revised discussion around Equation 2) – in other words, there is essentially a step-change in the threshold pressure between the tip and the side walls. The pressure can eventually exceed this new, higher threshold pressure at some point upstream if the finger is long enough, which is precisely the reasoning that leads to Δr_b (see revised discussion around Equation 4). This same reasoning suggests that viscosity will begin contributing to the finger width when Δr_b is itself comparable to $\sim 2R$, but branching will occur much earlier than that. The

rate-independence of the finger width in the “frictional regime” is also well established in previous work on viscously *unstable* frictional flow ([15], [19], [35]).

See also the next response, below.

Later on, on page 14, you state that “The finger width is independent of D_{visc} for individual fingers but” decreases with D_{visc} as self-confinement increases. But both are function of σ_0 , so how can that be?

We agree with the reviewer that this point was presented in a confusing way in the original manuscript. In the revised manuscript, we now present the pattern characteristics in Fig 8d-f by plotting against the strength of viscous forces ηQ , rather than against D_{visc} , which depends on ϕ . The associated description of the finger width now reads:

*“The finger width $2R$ (Fig. 8f) is rate-independent in the frictional fingering regime ($2R_{\text{fr}}$) is given by Eq. 1). Naively, one would expect increased viscous pressure within the finger to expand the width; instead, the fingers are observed to narrow when approaching D_{visc}^{**} . This narrowing is most likely a result of self-confinement, in which the competition between numerous fingers increasingly suppresses lateral expansion. There is a gradual transition from multiple individual fingers to side-by-side radial spokes as viscous stabilisation becomes stronger and stronger, as evidenced by $2R$ beginning to decrease before the system reaches D_{visc}^{**} . Note that the self-confinement effect is not included in the model which assumes constant $2R$ up to D_{visc}^{**} .”*

Also, why not do experiments at much smaller ϕ in figure 3 to confirm relationship (2): the range of ϕ over which the relationship is fitted is not convincing enough.

We sympathise with the temptation to, say, plot ϕ on a log scale and explore much smaller values. Physically, however, this quantity measures the thickness h of the bead layer relative to the thickness b of the gap. There is also a third length scale, the bead diameter d , that must be much smaller than the other two in order to treat the granular material as a continuum ($d \ll h < b$). There are several additional physical and practical limitations on these values. The thickness of the gap must be below the capillary length (a few mm) – we use $b = 0.9$ mm. The beads must be large enough that gravity overwhelms static electricity and other complicating effects – we use $d = 87 \mu\text{m}$. Practically, the bead layer is prepared by scraping with a straight edge that is supported on either side by strips that set the height. The resulting variability in ϕ is at least one grain diameter. To minimise the role of this variability, which can lead to furrows and general patchiness, the layer needs to be at least several grains thick – we use $h \sim 4d$. As a result, the minimum value of ϕ that is practically achievable with acceptable accuracy in our system is about $4d/b \sim 0.4$,

which is the smallest value considered in our study. Another practical complication of small ϕ is that the meniscus needs to travel much further in order to accumulate enough grains to bridge the gap (*i.e.*, $2R$ gets larger), eventually requiring a larger overall system size. For $\phi \sim 1$, in contrast, this accumulation happens very quickly and the system transitions to a fracture-like invasion pattern in which the finger width becomes less well defined as $2R$ approaches d .

To better communicate these limitations, we have revised the paragraph in the Methods section that describes the preparation of the granular layer. The paragraph now reads:

“Preparation of the granular layer: The dry hydrophobic beads were spread out on one of the treated glass plates (later to form the bottom surface of the Hele-Shaw cell). In order to achieve a granular layer of uniform thickness, two strips of adhesive tape were placed along opposite sides of the bottom plate, and a straight-edged tool resting on both tape strips was used to scrape the granular material into a uniform layer along the plate. The top plate was then mounted on top, separated from the bottom plate with 0.9 mm spacers. We varied the packing height by changing the thickness of the tape strips. Each strip consisted of several layers of tape film attached on top of one another. The tape film thickness was $63.5 \mu\text{m}$, and between 6 and 12 layers were used to create strips producing granular layer heights h between 0.38 and 0.76 mm, corresponding to ϕ between 0.42 ± 0.01 and 0.84 ± 0.01 , with the values verified and uncertainty estimated from trials where layers were made and then the mass measured independently. Experiments with lower filling fraction ($\phi = 0.21$) are shown for formation of spoke patterns, but note that filling fractions below 0.42 were not included in quantitative analysis of finger widths because of practical problems achieving uniform layer thickness for the thinnest layers.”

To further confirm the relationship illustrated in Figure 3, we have instead added data for two additional values of Q . Note also that the variation of R with ϕ has been discussed at length in previous work on viscously *unstable* frictional fingering (*e.g.* [15], [19]). Lastly, see also our response to comment 3 of Reviewer 3.

The relatively simple model used in numerical study might be ok, but I am really perplexed by the pressure scale obtained using it and reported in Figure 5 (and also a relatively low pressure used for fitting (7)). I find this really surprising having had experience with these system, so I would have expected to see some pressure measurements in experiments. I would strongly recommend that the authors do so, at least for some of the patterns. Simple back of the envelope calculations (using viscous fluid invading air) suggest that the pressure should be higher if I am not mistaken.

We agree with the reviewer that pressure measurements can provide helpful additional insight into displacement processes. However, we disagree with the reviewer's intuition about the magnitude of the pressure in our system.

Previous Figure 5 (now Figure 6) shows a simulation of water injection ($\eta = 10^{-3}$ Pa-s) at $Q = 100$ mL/min for $\phi = 0.42$ (this information is now included in the caption). The figure shows about 10 simultaneously growing fingers. If the injection rate is equally distributed amongst these fingers, then the flow rate through each finger is $Q/N \sim 10$ mL/min. The finger width for this value of ϕ is $R \sim 0.04$ cm. The cell has radius 13.4 cm and the fingers have nearly reached the edge in this snapshot. Taking a conservative finger length of $x \sim 15$ cm to account for meandering, Darcy's law suggests that $\Delta P \sim 6\eta Qx/(NRb^3) \sim 50$ Pa. If the magnitude of the threshold pressure at the tip (capillarity plus frictional stress) is comparable, then the magnitude of the pressure is ~ 100 Pa.

Pressure measurements in our experiments are consistent with the estimate above. Unfortunately, however, the pressure sensor in our system was attached to the tubing outside the cell, where the pressure signal is dominated by the flow through the tubing and in the inlet and entrance region of the flow cell, masking pressure fluctuations due to pattern formation within the flow cell. We have opted not to include this data, as it doesn't provide any new insight.

The model also systematically under-predicts the number of fingers in Figure 4 (at least for higher Q), but that is not commented on or discussed anywhere.

We believe the main shortcoming of the simulation (to be addressed in future work) is that the friction model is 1D and does not account for tangential stresses within the front, through which the friction depends on curvature. As a result, the selected fitting parameters represent a compromise between capturing the finger width and capturing the breakout of new fingers; the under-prediction of N at high Q is a consequence of this compromise. To better explain this trade-off in the manuscript, we have added more details at various points, including in Section II C:

“Note that we neglect the viscosity of the defending fluid, and that we use two fixed fitting parameters, the friction coefficient and a viscous coefficient, both set to match finger widths and viscous stabilisation across the range of experimental parameters (see Methods).”

And, also:

“Figure 2b shows simulation results across a range of filling fractions and injection rates, reproducing the experimental transition from single-finger to multi-finger growth as a function of Ca , although somewhat under-predicting the number of fingers at high Ca . Figure 3 shows the simulated finger width in the rate-independent ‘frictional regime’ at low Ca . Note that the

simulation uses an exponential friction model regardless of the curvature of the front, which gives a somewhat steeper decrease in $R(\varphi)$ compared to experiments and Eq.~(1)."

We have also added the following explanation to the Methods section:

"The simulated patterns match the experiments reasonably well across a wide range of ηQ and φ . A shortcoming of the current version of the simulation is that the frictional resistance is calculated using Eq. 11 regardless of the curvature of the front. The experimental data shows that the friction at the tip is better fitted to a linear friction model, with the side-wall friction more adequately described by the exponential model. The use of the exponential model in the simulations prioritises the viscous stabilisation dynamics which is controlled by side-wall friction, while the φ -dependency of the finger width is more steeply decreasing in the simulation compared to what is observed in experiments. Future improvements to the simulation model would therefore involve a curvature dependent friction in the front."

Lastly, note that the previous Figure 4 has changed considerably in light of the new analysis related to φ and the corresponding simulation results are no longer included to avoid cluttering the figure. The comparison between experiments and simulations is now captured by Figures 2, 3, 7, and 8.

Finally, I am not sure about (11). How is the finger length 13.4cm measured? Based on the instantaneous pattern that you have in figure 7? But then the nature of that pattern would not change if it was measured earlier in experiment (and a different r was therefore obtained). In that case, the threshold (11) is somewhat meaningless, and, as you have pointed out yourselves, more gradual.

We agree that our derivation of Equations 11 and 12 in the original manuscript may have been confusing due to the various interrelated expressions involved. In the revised manuscript, we have modified and clarified this derivation and the associated discussion (around Equations 8 and 9). The transitional values D^* and D^{**} are the values of D_{visc} at which the nominal finger-branching distance Δr_b reaches the system size $r_{\text{cell}} = 13.4$ cm and the finger width $2R$, respectively. As a result, the former value depends on system size and the latter does not. The associated transitions are indeed quite gradual, but these transitional values give a useful indication of the location of those transition regions.

The analysis in previous Figures 7 and 8, now combined in the revised Figure 8, is based on the instantaneous pattern at the moment when it first reaches the edge of the system (*i.e.*, $r_{\text{cell}} = 13.4$ cm). The patterns would look different and the quantitative values of the metrics would presumably change if we were to use snapshots at an earlier time, but we would still expect to see a transition at the same value of D^{**} . Considering snapshots at a later time would require a larger system

size, for which the transitional value of D^* would also change, but we would expect that new value of D^* to be indicative of the location of the transition.

We completely agree with the reviewer that it would be fascinating to observe the evolution of a single experiment from one single finger to multiple fingers to space-filling radial spokes, and that this should be possible in a large enough system. However, doing so remains a topic for future work because it is not practical in our current experimental system. In this first study, we restrict ourselves to presenting the discovery of these patterns and providing a theoretical framework that illuminates the basic mechanisms at play.

Reviewer #3 (Remarks to the Author):

This paper looks at the competition of frictional and viscous forces when a fluid displaces a sedimented layer of particles in a confined geometry. In typical experiments, either the frictional or viscous limits of fingering are investigated, but this article investigates the transition from friction dominated to viscous dominated patterns when the invading fluid is both less and more viscous than the defending fluid. The result of these experiments is a suite of striking patterns that form, all accompanied by simulations that recapitulate the experimental results as well as theory that captures the essence of the physical effects. The text, while being a bit wordy at points, clearly lays out a physical intuition for the observed phenomena and shows how the transitions in pattern morphology occur when a viscous length scale becomes comparable to a frictional length and then the system size. This study nicely bridges the displacement of dry granular particles to the classic experimental system of viscous fingering.

Overall this is a high-quality, interesting, and thorough study. I have a few questions and comments for the authors, detailed below. I do have a concern about several images in different figures that appear to have been manipulated in some way (see Major comment 3), though they do not seem to impact the results shown. However, I do recommend this for publication after minor revisions.

Major comments:

1. In Section II the authors derive an expression for the finger width that is independent of the viscous stresses in the low flow rate/ Ca number regime. Later in Sections II B and D, this same expression for the fingers is used when discussing patterns where new fingers emerge. When fingering occurs there is (as stated by the authors) a pressure build-up in the invading fluid that becomes comparable to the threshold pressure used to derive Eqn. 2. Please justify why, in the cases where it appears the viscous stress is of the same order as the frictional stress, the viscous stress can still be ignored when determining the finger size.

As noted above in response to a comment from by Reviewer 2, the viscous pressure along a finger is due to viscous flow through the finger. This viscous contribution is largest at the injection point and drops linearly with flow along the finger until it vanishes at the tip, where the pressure must exactly match the threshold for motion if the finger is moving. This threshold pressure is a combination of the capillary pressure from the meniscus and the friction from the bulldozed granular material (see text above Equation 1). Liquid invasion into a dry capillary tube without grains is exactly analogous – the viscous pressure must be maximum at the inlet and must vanish at the interface, where the pressure must balance the capillary pressure. We have revised Figure 1 to more clearly illustrate this point, including a new panel (d) that is specifically focused on pressure.

Because the threshold pressure for fringer growth is set by capillarity and granular friction, we expect it to be independent of both rate and position. (Both capillarity and granular friction become rate dependent at very high rates, the former because of dynamic-contact-angle effects and the latter because of granular fluidisation; all of our experiments are well below these transitions.) If multiple fingers of different lengths grow simultaneously, then all will share the same injection pressure and the same threshold pressure at the tip. As a result, a shorter finger must have a steeper pressure gradient along its length than a longer one.

We observe experimentally that the finger width is set at the tip, where the fluid pressure just balances the threshold pressure (see Figure 1b-d). The fingers do not widen upstream of the tip, even though the viscous pressure increases in that direction, because the straight, static side walls are harder to move than the curved, advancing tip (see revised discussion around Equation 2) – in other words, there is essentially a step-change in the threshold pressure between the tip and the side walls. The pressure can eventually exceed this new, higher threshold pressure at some point upstream if the finger is long enough, which is precisely the reasoning that leads to Δr_b (see revised discussion around Equation 4). This same reasoning suggests that viscosity will begin contributing to the finger width when Δr_b is itself comparable to $\sim 2R$, but branching will occur much earlier than that. The rate-independence of the finger width in the “frictional regime” is also well established in previous work on viscously *unstable* frictional flow ([15], [19], [35]).

Our experimental results reveal one other mechanism that influences finger width: the “self-confinement” effect, where neighbouring fingers suppress each other. This effect is not accounted for in our model.

To clarify these points, we have revised the last paragraph of Section II D, which now reads:

“The finger width $2R$ (Fig. 8f) is rate-independent in the frictional fingering regime ($2R_{\text{fr}}$ is given by Eq. 1). Naively, one would expect increased viscous pressure within the finger to expand the width; instead, the fingers are observed to narrow when approaching D_{visc}^ . This narrowing is most likely a result of self-confinement, in which the competition between numerous fingers increasingly suppresses lateral expansion. There is a gradual transition from multiple individual fingers to side-by-side radial spokes as viscous stabilisation becomes stronger and stronger, as evidenced by $2R$ beginning to decrease before the system reaches D_{visc}^* . Note that the self-confinement effect is not included in the model which assumes constant $2R$ up to D_{visc}^* .”*

2. There is a striking difference in the interface of the fingering patterns in the +M regime and the -M regime, notably that the +M regime is able to exhibit fluidization of the grains and not form a compacted layer of grains. Naively, I would assume that as

D_{visc} gets larger (particularly as G becomes small) that this could be achieved in the $-M$ case as well. Do you ever expect there to be a regime where there could be uniform, circular displacement ($c=1$, $s=1$) of the grains in the case where they are hydrophobic? If the $+M$ case could also fluidize the granular bed then it would be more clear that the transitions in the $\pm M$ cases are similar. Also, in this light, I hope the authors could comment on the importance of hydrophobicity versus hydrophilicity in the grains and if this is crucial for the observed physics.

We believe that it would indeed be possible to fluidise the grains in the $-M$ (viscously stable) regime. However, fluidisation occurs when viscous stresses in the fluid around and between the grains start to become comparable to contact stresses. We are far from this threshold in our viscously stable experiments because the defending phase is air, for which viscous effects will only become important at very high rates. As noted by the reviewer, we do see evidence of fluidisation at high rates in our viscously *unstable* experiments, but we have removed the viscously unstable results altogether for other reasons (see first response to Reviewer 2). This point suggests a promising avenue for future work.

The hydrophobicity of the grains is indeed central to our results. As noted in the Introduction, we have focused exclusively on drainage, where the invading fluid is nonwetting to the grains relative to the defending fluid. In this regime, the invading fluid bulldozes the defending mixture rather than invading the pore space of the packing. If the wettability were reversed (imbibition rather than drainage), the physics of fluid invasion would be entirely different because the invading fluid would preferentially invade the pore space and potentially even *pull the grains in* rather than *pushing them away*. We agree with the reviewer that this version of the problem is a fascinating avenue for future work. We now remind the reader at the end of the Discussion that:

“Multiphase frictional flows are thus a distinct class of fluid displacement problems in which pattern formation is controlled by capillarity, viscosity, and both inter-granular and sliding friction. Drainage at large capillary deformability and strong mobility ratio is now relatively well understood for both stable and unstable scenarios (e.g., Ref.~\cite{sandnes2011patterns} and the present study, respectively), but much of the parameter space remains unexplored.”

3. There are several instances where two panels in different figures are the same image, but are rotated mirror images of each other. The ones I noticed were: (i) Fig. 2a second row, the second image and Fig. 7c bottom row, second image, (ii) Fig. 6 a the top left image and Fig. 7a the lower left image, and (iii) four images of Fig. 7c and Fig. 9d the entire top row. I understand that you are showing the same experimental results in different contexts, but I am unsure why the images are manipulated in such a way, especially in showing a mirror image.

We apologise for the confusion. We intentionally re-used several experimental images across different figures for different purposes (*i.e.*, finger widths vs. number of fingers vs. pattern metrics). Unfortunately, some of those images were unintentionally reflected or rotated in the previous Figure 7 relative to the previous Figures 6 and 8. Earlier iterations of the previous Figure 7 had used the opposite axes (with Q on the vertical); to expedite transposing the axes, we at one point transposed the entire image, including the sub-panels, rather than rebuilding the figure from scratch. The experimental system is horizontal (with gravity perpendicular to page) and our metrics use the entire image, so the orientation of the images makes no difference to the physics or to the analysis. However, we regret the inconsistency and the potential confusion. In the revised manuscript, we have taken care that all images have the same orientation to facilitate cross-referencing between the different figures.

Minor comments:

1. There are a few places where it would be nice to have the numbers for parameters in the main text. In Section IIA when the experiment is introduced, please write the gap spacing. This is useful so that when the particle size and finger width are mentioned the reader has a sense of the relative size of the system. It also makes estimation of different pressures (P_b , P_t) more accessible for the reader to check.

We agree with the reviewer and now specify the gap spacing ($b = 0.9$ mm) in Section II A.

2. At two points in the text you have fit parameters that could have some physical interpretation: σ_0 on page 5 and ΔP_b on page 7. Does the value of σ_0 make sense with friction of dry grains?

The theoretical model now includes two fitting constants, σ_0 for the linear friction at the finger tips and σ_β for the exponential friction at the sidewalls. The former represents the friction stress per unit length b of the granular front at the tip. The front bridges the gap and σ_0 accounts for friction with both the top and bottom plates. We find a best fit value of $\sigma_0 = 16$ Pa; it is not clear how to relate this value to, for example, the parameter μ from the angle of repose in dry grains. The fitting constant σ_β in the exponential model represents the stress in the granular packing at the far side of the front. We have now clarified and expanded our discussion of these models throughout Section II A.

The value of $\Delta P_b = 30$ Pa is very similar to P_t from Eqn. (1) if you calculate that quantity from the data in Fig. 3. If we interpret P_t as the pressure needed to move a compacted, static region of beads, then having that same pressure difference (ΔP_b) away from the tip of a finger cause new growth seems reasonable. Or is there a different interpretation for this?

The reviewer makes an interesting point. In the revised manuscript, we have expanded the scope of the model to additionally account for the role of ϕ and the fundamental idea for ΔP_b is very similar to what the reviewer suggests here. We find that the wall friction increases exponentially with front thickness L , as illustrated in the inset of Figure 5b. However, we retain a linear friction model for P_t , which sets the finger width, for simplicity and to account for the lower frictional resistance associated with a curved front. This model (Equation 3) agrees well with experimental results across the range of parameters studied here.

3. On line 391 you mention that conservation of mass gives you another estimate of ϕ , but this expression assumes that the packing fraction of the undisturbed bed and the compacted region of grains are the same. The density of random granular packings can be sensitive to their method of preparation, is there evidence that the two regions have the same packing fraction?

We apologise for the confusion. The text describing the estimation of ϕ was erroneously included as a fragment from a much earlier draft. In fact, we do not estimate ϕ from the images because this method is sensitive to somewhat arbitrary thresholding choices. In addition, we agree with the reviewer that the packing fraction might be different in these two regions. Instead, we use the much simpler approach of estimating ϕ from the height of the strips used to prepare the layer. The relevant text in the method section now reads:

“Preparation of the granular layer: The dry hydrophobic beads were spread out on one of the treated glass plates (later to form the bottom surface of the Hele-Shaw cell). In order to achieve a granular layer of uniform thickness, two strips of adhesive tape were placed along opposite sides of the bottom plate, and a straight-edged tool resting on both tape strips was used to scrape the granular material into a uniform layer along the plate. The top plate was then mounted on top, separated from the bottom plate with 0.9 mm spacers. We varied the packing height by changing the thickness of the tape strips. Each strip consisted of several layers of tape film attached on top of one another. The tape film thickness was 63.5 μm , and between 6 and 12 layers were used to create strips producing granular layer heights h between 0.38 and 0.76 mm, corresponding to ϕ between 0.42 ± 0.01 and 0.84 ± 0.01 , with the values verified and uncertainty estimated from trials where layers were made and then the mass measured independently. Experiments with lower filling fraction ($\phi = 0.21$) are shown for formation of spoke patterns, but note that filling fractions below 0.42 were not included in quantitative analysis because of practical problems achieving uniform layer thickness for the thinnest layers. The cell was clamped together firmly after assembly to prevent the top plate from lifting. All four edges of the cell were left open to the atmosphere.”

4. For Eqn. 16 a fit value of $C=8$ is used, is there some justification for having this be a fit parameter instead of using $C=6$ derived from Poiseuille flow in a Hele-Shaw

cell? Also, I'm concerned about the units of Eqn. 16, if C is dimensionless then X also needs to be so: is X normalized by the gap length?

The viscous response in the simulations was calibrated to give a good match between experiments and simulations across the full range of parameters studied here. The resulting calibration suggested a factor of 1.33 relative to the expected expression for Poiseuille flow in a Hele-Shaw cell, which led to $C = 1.33 \cdot 6 = 8$ in Equation 6 of the original manuscript. For clarity, we have separated the factor of 6 from the calibration constant in the revised manuscript, so the coefficient is now $6C$ with $C = 1.33$ (now Equation 12).

A value $C > 1$ indicates a higher viscous resistance than ideal Hele-Shaw flow. One obvious reason for this higher resistance is that our fingers are not ideal Hele-Shaw cells; rather, each finger is essentially a meandering, rough-walled rectangular channel. The aspect ratio of each finger, $2R/b$, ranges from about 5 to 10 (Figure 3). Over this range, even a straight, smooth-walled rectangular channel has a resistance of 5-30% higher than an ideal Hele-Shaw cell. As a result, we believe that $C = 1.33$ is entirely reasonable.

There is still some degree of disagreement between the simulations and the experiments. We believe that the main reason for these differences is that the simulation uses an exponential model for friction everywhere, whereas the experiments suggest that the friction should weaken with the curvature of the front. We comment on this source of error in the Methods, last paragraph of Section IV B.

Lastly, we believe that X should indeed have dimensions of length (not normalized). Recall that the quantities in this equation (now Equation 12) are *pressures*, not *pressure gradients*.

5. The authors have included fantastic videos as supplementary material, it may be nice to actively reference them in the text so the readers are more aware of their existence.

We thank the reviewer for this excellent suggestion. We have now included references to the Supplementary Movies at appropriate places in the main text and in the captions of Figures 2 and 7. Note that Supplementary Movies 4-7, which showed viscously *unstable* results, have now been removed since that regime is no longer included in the manuscript.

REVIEWER COMMENTS

Reviewer #1 (Remarks to the Author):

In this revised paper the authors have done a good job better introducing the physical problem, the rearrangements possible in the granular material during fluid invasion, and presenting various aspects of the comparisons of experiments and simulations. I can better appreciate the novelty of the work presented and the challenge of understanding and modelling the complex multiphase situation. As given below in the various comments, especially as I do not work in the field but had to discover myself by looking at some of the references, I do find it 1) frustrating that some of the model equations presented as apparently new to this paper have already been given in a previous paper by the authors on this basic physical problem (so there is some literature, by the authors and others, on this kind of problem, which is somewhat hidden in the introduction), 2) unclear and frustrating that the authors refer, sometimes in the same sentence, as the simulations agreeing with the data and having significant variability, and 3) various summary graphs, or phase diagrams should really be given in, or perhaps have added, dimensionless forms so a reader can better judge the self-consistency of all the arguments. While I can better appreciate the work I can't help but think that a further revision is needed to properly present the findings to the community.

Other remarks:

1) Figure 1d label: what is N in Q/N ? We seem to only learn what is N on p. 8.

2) p. 6: What does the "Janssen coefficient" actually represent and what are its dimensions? A reader does not have to go look up the paper referenced

3) The authors should tell the reader that equation (1) was previously given in reference 42, equation (8). In the same vein the authors should inform a reader that equation (2) was previously given in reference 42, equation (5).

4) p. 7: "...be described as a frictional regime in which the dynamics of finger growth are determined by the local competition between capillary and frictional forces at the advancing finger tip, with viscous effects playing no role." Isn't this an exaggeration that "with viscous effects playing no role" since figure 3 clearly shows some effects of flow rate.

a) Moreover, in the concluding remarks in the discussion we find “Pattern formation in this ‘frictional regime’ is rate independent.” But isn’t this statement inconsistent with the Q dependence evident in Figure 3?

5) Top of p. 9: “we hypothesize an exponential friction law of the form” – but it seems to be true, as stated by the authors in their 2018 Physical Review Fluids paper, unless I am confusing concepts, that this idea has already been recognized in the literature for this class of problems, as they write in 2018: “this yield stress has previously, in the context of labyrinth patterns [29] and of plug formations in narrow tubes [41], been assumed to be exponentially increasing in front thickness L. The exponential behavior can be justified by considering Janssen’s model for stresses in packings of grains, which assumes a linear relationship between the principal stresses in the packing, in conjunction with the static Coulomb frictional stresses at the plate boundaries of the cell.”

6) p. 10: “Figure 5c shows a reasonable collapse of all N data against ... although there is considerable variability” – you can’t have it both ways. Either there is “reasonable collapse” or there is “considerable variability”. In my view, inspection of the data in figure 5c shows considerable variability, even when Q is fixed. I realize the system is complex but the language is ambiguous and unfortunately seems to hide whether there is really good agreement or just apparent or little agreement. In fact, in figure 5c the horizontal range of variation is at least a factor of 3 for any sets of data (same vertical value), which is also the range of variability of $(1-\phi)/\phi$ (based say on ϕ values in figure 2), so it is unclear to me whether it is actually true that “our simple model captures reasonably well the main effects of viscous stabilisation and the role of ϕ in suppressing the sprouting of new fingers” (though I agree on physical grounds that increasing ϕ can have an effect but that is not new to this paper).

7) Although the introduction introduces the authors suggestion/claim about their use of D_{visc} it is only on p. 14 that we realize, apparently, that similar ideas have already been introduced by at least two different research groups as well as a 2017 paper from some of the current authors.

8) p. 14: “we plot η^* as a function of Q in the phase diagrams (blue line)” – What is η^* ?

9) Since the authors have a detailed theory containing all of the experimental parameters, figures 8d-f would be better with the horizontal axis dimensionless rather than dimensional ($\eta_{\text{inv}} Q$) so as to better appreciate how well the simulations/theory represent the data.

10) Vagueness continues in the discussion where we find: “The estimated transitions

from single to multiple fingers (Q) and from multiple fingers to spoke pattern (Q) (see Eqs. (24) and (25) in the Supplementary Information) are plotted in blue and orange, respectively, and agree reasonably well with the visual characteristics of the patterns, although

the transitions are gradual and there is a fairly large degree of variability in the system." To my eye the plot is not so convincing, particularly the "boundary" suggested by the orange curve. In any event, a better plot would be dimensionless, since (again) the authors have a model apparently connecting all of the variables, e.g., $\eta_{\text{inv}} Q$ and ϕ .

Reviewer #2 (Remarks to the Author):

This is my second review of the paper. I appreciate changes made by the authors, but still have some reservations.

(1) Regarding my comment about the pressure - I agree with the estimates regarding the pressure drop along the finger, but my original comment was regarding the magnitude of pressure overall, e.g. at $r=0$ in figure 6 your report 100 Pa. I don't see why this is not measurable using your method - granted, there are some line losses that you need to account for due to tubing etc, but surely you know what they are.

(2) The statement "with increased forcing" in the abstract is confusing.

(3) In section 2A you define the log of viscosity ratio - not sure what for, given that the defending fluid is inviscid and remains the same throughout the paper. You also introduce a finger width with \overline{R} , but refer to it with just R elsewhere, and R is properly defined only in caption to Fig. 3 - it should be clear what it is immediately. Out of curiosity, and given that you haven't really defined R_t : is $R=2R_t$ (can you show the corresponding circle in one of your experimental images)?

(4) I am not too keen on pre-empting results by a statement "At low Q (and therefore low Ca), viscous forces within the fingers are negligible; surface tension and friction compete to set the finger width $2R$." I think that this should come after you have presented evidence for it. Also, detailing Janssen law on page 6 long before you use it is not ideal, since you repeat yourself later. There is "were" missing in "except where new fingers initiated" on page 5.

(5) Please clarify statement “viscous stabilisation” when you use it for the first time - you refer to the fingers growing radially and filling space, but you still get fingering.

(6) This statement is confusing “which is consistent with the fluid pressure along a finger increasing with the length of a finger and with the flow rate through it.”

This can't be correct, because there needs to be a pressure gradient for finger to grow. What you mean is that absolute pressure increases overall for increasing values of flux etc Be sure to say it.

(7) Could you add points from Fig. 5a to the inset of Fig.5b as well?

(8) There is some confusion with the use of D^* and D^{**} . I don't understand how blue and orange lines in Fig. 8(a) and (b) are straight, shouldn't the boundaries be ν^{-1}/Q as per definition of D . Also, not sure if you should quote ν^{**}/ν^* - you means some ratio of prefactors only. In Fig.8 you use D and ν/Q interchangeably - you should be more careful. In caption to the figure you refer to a disk diameter - what disk?

(9) S_{front} is not clearly defined - what is the "outer perimeter of the connected advancing front"? Given that the from is still fingered, that ought to be defined somewhere.

Reviewer #3 (Remarks to the Author):

The new manuscript has removed the viscously unstable situation from the previous revision and has included additional explanation for the viscously stable regime, this has led to a version of the manuscript that contextualizes the work and explains the effect more clearly. The authors have addressed concerns and comments in their response well and I recommend this manuscript for publication.

One minor comment: for the inset of Figure 5b there are no axes values. I appreciate that this is just to demonstrate the exponential relationship of the data, but I think there should still be values on the axes to make the relationship clear.

We thank the reviewers for their constructive feedback. In the responses below, the comments of the reviewers are included verbatim in black and our replies follow in blue.

Reviewer #1 (Remarks to the Author):

In this revised paper the authors have done a good job better introducing the physical problem, the rearrangements possible in the granular material during fluid invasion, and presenting various aspects of the comparisons of experiments and simulations. I can better appreciate the novelty of the work presented and the challenge of understanding and modelling the complex multiphase situation.

We thank the reviewer for this positive assessment, and again for their initial feedback that prompted the previous revision.

As given below in the various comments, especially as I do not work in the field but had to discover myself by looking at some of the references, I do find it

1) frustrating that some of the model equations presented as apparently new to this paper have already been given in a previous paper by the authors on this basic physical problem (so there is some literature, by the authors and others, on this kind of problem, which is somewhat hidden in the introduction),

We certainly agree with the reviewer that the precise novelty of this work must be clear and unambiguous to the reader. However, we disagree that anything was “somewhat hidden” in the introduction. The second paragraph of the introduction is entirely about previous work on viscously *unstable* frictional fingering, which has indeed been studied fairly extensively by BS and co-workers. References to that work are also included throughout the manuscript in relevant places.

The contribution of the present study is to extend and adapt this understanding to viscously *stable* frictional fingering, which, as highlighted in the last paragraph of the introduction, is an unstudied problem where fundamentally different behaviour is expected. In the revised manuscript, we have made a variety of minor changes to make links with previous work even more explicit. These changes are explained in more detail below.

2) unclear and frustrating that the authors refer, sometimes in the same sentence, as the simulations agreeing with the data and having significant variability, and

As detailed below, we have made various minor changes to be more clear and specific about the senses in which the simulations and the data agree vs. disagree.

3) various summary graphs, or phase diagrams should really be given in, or perhaps have added, dimensionless forms so a reader can better judge the self-consistency of all the arguments. While I can better appreciate the work I can't help but think that a further revision is needed to properly present the findings to the community.

We thank the reviewer for insisting on this point; ultimately, we agree. As detailed below, we have made several minor changes to the presentation of our results in order to put dimensionless quantities at the forefront where appropriate.

Other remarks:

1) Figure 1d label: what is N in Q/N? We seem to only learn what is N on p. 8.

We agree with the reviewer that N and Q/N are not yet relevant at the point where Figure 1 is introduced and discussed. Thus, we have removed the label “Q/N” from Fig. 1d.

2) p. 6: What does the “Janssen coefficient” actually represent and what are its dimensions? A reader does not have to go look up the paper referenced

We agree with the reviewer. To clarify the basic definition of the Janssen coefficient for our system, we have added the following statement, from which it can also be inferred that the Janssen coefficient is dimensionless: “...and κ is the ratio of out-of-plane normal stress to in-plane normal stress in the granular packing (i.e., the Janssen coefficient) [39,41]”.

3) The authors should tell the reader that equation (1) was previously given in reference 42, equation (8). In the same vein the authors should inform a reader that equation (2) was previously given in reference 42, equation (5).

Indeed, this model for finger width (or a very similar one) was derived in Ref [42] for viscously *unstable* frictional flows. The same arguments are relevant here for *viscously stable* frictional flows when viscous forces are sufficiently weak (i.e., in what we now refer to as the rate-independent regime). To make this link more explicit, we have thoroughly revised the section describing this derivation. For example:

- The paragraph above Equation 1 now opens with the sentence “Following previous work [15,42,43], we now estimate the characteristic rate-independent finger width $2R_f$ that balances capillarity with friction by seeking the value of R_f that minimises [the total yield pressure at the tip] P_t .”
- Equation 1 itself is now followed by the sentences “This finger width $2R_f$ is thus the emergent natural length scale in our system. A very similar expression has been derived and used for viscously unstable frictional fingers [15,42,43].”
- We have demoted the former Eq. 2 to an in-line expression and added a citation to Ref. [15] at the end of that sentence.

4) p. 7: “...be described as a frictional regime in which the dynamics of finger growth are determined by the local competition between capillary and frictional forces at the advancing finger tip, with viscous effects playing no role.” Isn’t this an exaggeration that “with viscous effects playing no role” since figure 3 clearly shows some effects of flow rate.

Upon further consideration, we agree with the reviewer that the term “frictional regime” is inappropriate, as discussed below. However, we disagree about the impact of flow rate. Figure 3 plots the finger width $2R$ against the filling friction ϕ for different flow rates Q . The results exhibit some degree of natural variability or “noise”, as should be expected in any pattern-forming system and in any frictional system. However, there is a distinct decreasing

trend with ϕ that is nicely captured by Equation 1, whereas there is no clear trend with Q that we can see. This is consistent with previous work on viscously *unstable* frictional fingering, where the finger width was found to be independent of Q at low injection rates. Figure 8e also supports the idea that $2R$ is independent of D_{visc} for small D_{visc} , deviating from $2R_f$ as D_{visc} increases.

Separately, the term “frictional regime” implies that frictional forces are somehow more important at low Q , which is not the case. Frictional forces are always important in our system because the grains are submerged in a low viscosity fluid (air), so the bulldozing front remains ‘frictional’ throughout the range of flow rates studied here. This is in contrast with the viscously *unstable* system (e.g., Sandnes et al. 2011), where the grains are submerged in a high-viscosity fluid and can be fluidised at high Q , transitioning to what was termed the ‘viscous regime’ in previous work. Hence, the terms ‘frictional regime’ and ‘viscous regime’ have clear meanings in the viscously *unstable* context that do not extend to the present (viscously *stable*) context. To avoid these terms, we have removed the sentence highlighted by the reviewer (formerly on pg. 7) and we have replaced the term ‘frictional regime’ with the term ‘rate-independent regime’ throughout the manuscript (references to the viscous regime were removed in the previous revision).

a) Moreover, in the concluding remarks in the discussion we find “Pattern formation in this ‘frictional regime’ is rate independent.” But isn’t this statement inconsistent with the Q dependence evident in Figure 3?

As also noted in that response, we do not believe that Figure 3 shows any dependence of $2R$ on Q .

5) Top of p. 9: “we hypothesize an exponential friction law of the form” – but it seems to be true, as stated by the authors in their 2018 Physical Review Fluids paper, unless I am confusing concepts, that this idea has already been recognized in the literature for this class of problems, as they write in 2018: “this yield stress has previously, in the context of labyrinth patterns [29] and of plug formations in narrow tubes [41], been assumed to be exponentially increasing in front thickness L . The exponential behavior can be justified by considering Janssen’s model for stresses in packings of grains, which assumes a linear relationship between the principal stresses in the packing, in conjunction with the static Coulomb frictional stresses at the plate boundaries of the cell.”

We completely agree with the referee that this exponential friction law is not inherently new. Indeed, the exponential increase in wall friction for a granular material in a confined geometry is the classical Janssen effect. We had no intention of presenting this concept as a key or novel aspect of our work. To make this clear, we have revised the relevant sentence: “To capture this feature, we follow previous work [15,39,40] by hypothesising an exponential friction law of the form...”, where Ref [40] is the classical paper by Janssen.

6) p. 10: “Figure 5c shows a reasonable collapse of all N data against ... although there is considerable variability” – you can’t have it both ways. Either there is “reasonable collapse” or there is “considerable variability”. In my view, inspection of the data in figure 5c shows considerable variability, even when Q is fixed. I realize the system is complex but the language is ambiguous and unfortunately seems to hide whether there is really good

agreement or just apparent or little agreement. In fact, in figure 5c the horizontal range of variation is at least a factor of 3 for any sets of data (same vertical value), which is also the range of variability of $(1-\phi)/\phi$ (based say on ϕ values in figure 2), so it is unclear to me whether it is actually true that “our simple model captures reasonably well the main effects of viscous stabilisation and the role of fingers” in suppressing the sprouting of new (though I agree on physical grounds that increasing ϕ can have an effect but that is not new to this paper).

We agree with the reviewer that the results presented in Figure 5 could be strengthened, and that more precise language is merited around describing them. We have made several substantial changes to address these comments and some related comments by the other reviewers. In particular:

- We have elevated the inset of Figure 5b into a new panel 5c and added all available data sets to this plot. The result now more clearly illustrates the exponential trend between $\Delta\{P_b\}$ and $\sqrt{\phi/(1-\phi)}$. An additional benefit of this change is that all panels are now slightly larger.
- We have substantially revised our description and presentation of the model for N (around Eqs. 3-7) to clarify the role of D_{visc} . We now use a dimensionless quantity related to D_{visc} on the horizontal axis of Figure 5d (formerly 5c).
- We have substantially revised our description of the data presented in Figure 5 to clarify the application of the model to the results (second to last paragraph of the section).
- We have revised the last paragraph of the section, in which we discuss the results presented in Figure 5d (formerly 5c). We have removed the reference to “collapse” in favour of making the unambiguous point that the data exhibits a clear trend that is well captured by the model, while still exhibiting substantial variability around that trend. That paragraph now reads:

“The simple model captures the overall trend in the data, despite containing none of the geometrical complexity of the branching patterns. Note, however, that there is considerable variability in the data. The actual strength of viscous forces with a finger depends on the finger width, which exhibits 10-20% variability around the characteristic value R_f at the same value of ϕ (Fig. 3). On the vertical axis, we observe that N can vary both between different experiments and also within a single experiment as fingers start, stop, and sometimes restart again, making this measurement inherently imprecise. Note also that, for large N , the fingers begin to fill the available space, crowding out the formation of new fingers. This effect is not included in the model, but should suppress the growth of N at even higher values of $\sqrt{VD_{\text{visc}}/(R_{\text{fbr_out}})}$. Nevertheless, our simple model captures reasonably well the roles of viscosity and friction in controlling the sprouting of new fingers.”

Regarding the vertical variability in Figure 5d, note that our values of N exhibit a variability of roughly ± 2 fingers. As noted in the quote above, this quantity is inherently imprecise, but it is the best available metric by which to quantify the evolution of the pattern. Regarding the horizontal variability in Figure 5d, the data forms a band around the model with an overall width of 5 to 10 on the new dimensionless horizontal scale. The reviewer’s point that the impact of varying ϕ could get lost within this band is duly noted, but readily refuted by the clear trend with ϕ shown in Figures 3 and 5b. However, it is also clear that increasing the strength of viscous forces relative to friction is dominating the trend, providing strength for

our view that “our simple model captures reasonably well the roles of viscosity and friction in controlling the sprouting of new fingers”.

7) Although the introduction introduces the authors suggestion/claim about their use of D_{visc} it is only on p. 14 that we realize, apparently, that similar ideas have already been introduced by at least two different research groups as well as a 2017 paper from some of the current authors.

The fluid-driven deformation of a porous solid has been studied from a variety of different perspectives. In all of these scenarios, it is logical to construct a dimensionless number that compares the forces driving deformation (e.g., viscosity) to those resisting deformation (e.g., confining stress, yield stress, gravity, friction, etc); our D_{visc} is one such number. All of these dimensionless numbers are similar in spirit, but no more so than, say, a capillary number for a viscously unstable flow and a Bond number for a gravitationally unstable flow – both numbers compare the destabilising force (viscosity or gravity, respectively) to the stabilising force (capillarity in both cases). In any specific case, the relevant question is not about what forces should be compared, but rather about how the balance of those forces should be expressed. Clearly, the correct expression for the balance between viscosity and frictional bulldozing & bridging (Eq. 5) is both novel and non-obvious.

To clarify these points, we modified the first mention of D_{visc} (in the introduction) to be more specific: “[W]e show that the pattern formation is controlled by the strength of viscous forces in the invading phase relative to friction due to bulldozing and pile-up of grains in the defending phase, as quantified by a “viscous deformability” parameter D_{visc} .”

We have also made some minor changes to the paragraph mentioned by the reviewer, which is on pg. 13-14 and now reads:

“Our D_{visc} is similar in spirit to the large-capillary-number limit of the ‘fracturing number’ of Holtzman et al. [21], where the motion of a granular solid is resisted by friction under confining stress; to the ‘viscous fracturing number’ of Carrillo & Bourg [44], where the motion of a porous viscoplastic solid is resisted by a yield stress; and to the ‘fluidisation number’ of Campbell et al. [25], where the motion of a granular material was resisted by friction due to the weight of the grains. In the present context, friction is instead controlled by bulldozing, pile-up, and bridging.”

8) p. 14: “we plot η^* as a function of Q in the phase diagrams (blue line)” – What is η^* ?

We agree with the reviewer that the quantities η^* and η^{**} might have been confusing. We have now removed these quantities in favour of D_{visc}^* and D_{visc}^{**} : “We indicate these transitional values [D_{visc}^* and D_{visc}^{**}] in the η - Q phase diagrams (diagonal blue and orange lines, respectively in Figs. 8a and b) for fixed $\phi=0.49$, $b=0.9\text{mm}$, and $r_{\text{out}}=13.4\text{cm}$, for which $D_{\text{visc}}^{**}=860$.”

9) Since the authors have a detailed theory containing all of the experimental parameters, figures 8d-f would be better with the horizontal axis dimensionless rather than dimensional ($\eta_{\text{inv}} Q$) so as to better appreciate how well the simulations/theory represent the data.

We thank the reviewer for the suggestion. In the previous revision, we were using a dynamic definition of D_{visc} that varied with time (via the total injected volume) and with the number of active fingers. This definition is useful because the strength of viscous forces decreases when new fingers form (partitioning the flow rate) and increases as the fingers grow longer. Such a definition is analogous to the “local” capillary numbers that are sometimes used in the literature on viscous fingering, and which are similarly time- and pattern-dependent. However, this definition made it awkward to plot patterns and pattern characteristics against D_{visc} (such as in Figures 8 and 9). We have now reverted to a static definition of D_{visc} that depends only on fixed geometric parameters and material properties. Courtesy of this change, we now use D_{visc} on the horizontal axis of Figures 8d-f, on the horizontal axis of Figure 5d (see response to point 6), and on the vertical axis of Figure 9 (see next point).

10) Vagueness continues in the discussion where we find: “The estimated transitions from single to multiple fingers ($\eta^{-1} Q$) and from multiple fingers to spoke pattern (ϕ) (see Eqs. (24) and (25) in the Supplementary Information) are plotted in blue and orange, respectively, and agree reasonably well with the visual characteristics of the patterns, although the transitions are gradual and there is a fairly large degree of variability in the system.” To my eye the plot is not so convincing, particularly the “boundary” suggested by the orange curve. In any event, a better plot would be dimensionless, since (again) the authors have a model apparently connecting all of the variables, e.g., $\eta^{-1} Q$ and ϕ .

We thank the reviewer for suggesting that we replot this figure (Figure 9) in terms of D_{visc} . As discussed in response to the previous point, our previous dynamic definition of D_{visc} made it awkward to do so; however, we have now adopted a static definition of D_{visc} that makes this much more straightforward. Hence, we now use D_{visc} on the vertical axis of Figure 9.

We have also rephrased the offending sentence in the discussion to be more clear about the meaning of the orange and blue curves in Figures 8 and 9: *“Both transitions are gradual and the system exhibits a fairly large degree of variability, so neither value of D_{visc} represents a sharp phase boundary; rather, they are indicative of the expected location of the transition region. As such, these transitional values agree reasonably well with the evolution of the geometric features (Fig. 8d-e) and visual characteristics (Fig. 9) of the patterns.”*

Reviewer #2 (Remarks to the Author):

This is my second review of the paper. I appreciate changes made by the authors, but still have some reservations.

(1) Regarding my comment about the pressure - I agree with the estimates regarding the pressure drop along the finger, but my original comment was regarding the magnitude of pressure overall, e.g. at $r=0$ in figure 6 your report 100 Pa. I don't see why this is not measurable using your method - granted, there are some line losses that you need to account for due to tubing etc, but surely you know what they are.

We agree with the merits of comparing pressure at the central node of the simulation to pressure measured at the inlet in the experiments. We have limited data available, but are

able to make a direct comparison of pressure vs time for $D_{\text{visc}} = 31$ in the new Figure 6b (closely matching Fig 6a where $D_{\text{visc}} = 22$). The comparison demonstrates that the overall fluid pressure in the simulations is comparable to the experiments, and that the fluctuations in pressure caused by the evolution of the pattern are well matched.

(2) The statement “with increased forcing” in the abstract is confusing.

We agree with the reviewer that this phrase was ambiguous. The sentence now reads: “...where we observe a transition from growth of a single frictional finger to simultaneous growth of multiple fingers as viscous forces are increased.”

(3) In section 2A you define the log of viscosity ratio - not sure what for, given that the defending fluid is inviscid and remains the same throughout the paper.

We agree that the importance of the log viscosity ratio is now much reduced, having removed the viscously *unstable* results. However, we have elected to retain it as a touchstone that may be useful in the context of future work (specifically, the idea that this study focuses on “strongly negative M ”, indicating a viscously *stable* flow in which the viscosity of the defending fluid is unimportant).

You also introduce a finger width with \overline{R} , but refer to it with just R elsewhere, and R is properly defined only in caption to Fig. 3 - it should be clear what it is immediately. Out of curiosity, and given that you haven't really defined R_t : is $R = 2R_t$ (can you show the corresponding circle in one of your experimental images)?

Our original intention was that \overline{R} was a characteristic fixed value of the finger width ($2\overline{R} = 1\text{cm}$) that was exclusively used to define the capillary number Ca in such a way that the value was known a priori, without reference to experimental outcomes. To avoid confusion, we have now removed both \overline{R} and Ca . We have replaced Ca with Q in the earlier parts of the manuscript, where results are primarily dimensional and where Q was ultimately the quantity being varied; we now focus more strongly on D_{visc} in the later parts of the manuscript.

The quantity $2R$ is the actual measured finger width in experiments and simulations. This quantity is introduced in the second paragraph of SII.A, illustrated in Figure 1b and 1c, and plotted in Figure 3. To elaborate on the measurement of R , we have added the following sentence to the end of the third paragraph in SII.A: “Values for R were obtained from final images (such as those presented in Figure 2) by measuring the ratio of fluid-filled invaded area to internal finger interface length $R = A_{\text{fluid}}/S_{\text{finger}}$.”

We have now also introduced a new quantity $2R_f$ that is the characteristic finger width in the rate-independent regime, as presented in Eq. 1 and plotted in Figure 3 (solid curve). The quantity $2R$ is the actual finger width observed in experiments and simulations, as also plotted in Figure 3 (symbols). The quantity R_t is the radius of curvature at the tip of a finger, as illustrated in Figure 1c and assumed here to be constant. R_t is ultimately only used within the derivation of R_f . The theory predicts that $R_f = 2R_t$, but this is a modelling result rather than a definition. We have revised the three paragraphs above Equation 1 to clarify the different roles of R_t and R_f .

(4) I am not too keen on pre-emptying results by a statement “At low Q (and therefore low Ca), viscous forces within the fingers are negligible; surface tension and friction compete to set the finger width $2R$.” I think that this should come after you have presented evidence for it.

We agree with the reviewer. We have revised this statement and integrated it into the discussion of Figure 3, which now reads: “We plot $2R$ as a function of ϕ in Figure 3 for experiments and simulations at low rates Q , comparing against $2R_f(\phi)$ from Eq. 1. The theory agrees well with the experimental observations, in common with previous research on viscously unstable air-invasion labyrinths [15,42] and confirming the expectation that there exists a rate-independent regime where viscous forces within the fingers are indeed negligible, such that capillarity and friction compete to set the finger width.”

Also, detailing Janssen law on page 6 long before you use it is not ideal, since you repeat yourself later.

We agree that the repetition of the Janssen argument is not ideal, but we have elected to retain it because this repetition is necessary in order to properly introduce the characteristics of the friction models and the differences between side-wall and fingertip friction.

There is “were” missing in “except where new fingers initiated” on page 5.

Corrected with thanks.

(5) Please clarify statement “viscous stabilisation” when you use it for the first time - you refer to the fingers growing radially and filling space, but you still get fingering.

Our first use of the term “viscous stabilisation” is in the introduction, in reference to fluid-fluid displacement in Hele-Shaw cells and rigid porous media: “Without the grains, reversing the two viscosities ($\eta_{def} < \eta_{inv}$) negates the fingering instability by turning viscosity into a stabilising force. With grains that are fixed in place, capillary forces and pore-scale disorder compete with viscous stabilisation to produce fractal invasion-percolation patterns at low injection rates and rough but stable fronts at high injection rates.” Thus, it is clear that viscous stabilisation does not preclude the formation of patterns.

The term is next used in SII.B: “A sufficiently high pressure along the length of a finger can drive new fingers to break out from the side walls, leading to growth in the central parts of the pattern; this is the manifestation of viscous stabilisation in this frictionally unstable system.” Note that we have revised the latter half of this sentence to make it clear that, in this system, viscous stabilisation leads to a more compact pattern by driving the creation of more fingers rather than fewer. To further emphasise this point, we have also added a sentence to the end of the first paragraph of SII.B: “Thus, strikingly, viscous forces drive the pattern to be more space-filling in this system by promoting the formation of more fingers.”

The term “viscous stabilisation” is then used in various places throughout the manuscript, including in SII.D in reference to spoke patterns: “... producing extreme viscous stabilisation and fingers that radiate outward in an axisymmetric spoke pattern. Here, the frictional

instability produces fingers and viscous stabilisation forces them to grow radially, with an axisymmetric viscous pressure field. The tips remain equidistant from the injection point, creating a circular displacement front with embedded radial streaks of packed grains. As the pattern expands over time, the fingers increase in number by splitting to populate the growing circumference while maintaining a constant characteristic finger width (see Supplementary Movie 3)."

(6) This statement is confusing "which is consistent with the fluid pressure along a finger increasing with the length of a finger and with the flow rate through it."

This can't be correct, because there needs to be a pressure gradient for finger to grow. What you mean is that absolute pressure increases overall for increasing values of flux etc Be sure to say it.

We have slightly rephrased this statement for clarity: "which is consistent with the fact that the fluid pressure at any fixed radial position along a finger must increase with the length of that finger and with the flow rate through it."

(7) Could you add points from Fig. 5a to the inset of Fig.5b as well?

We thank the reviewer for this suggestion. We have now promoted this inset to a new panel Fig. 5c and included all available data. Re-analysis of this data has led to a slightly different value of σ_{β} : $\sigma_{\beta} = 0.20 \pm 0.02$, where the 95% confidence interval from the least-squares fitting is used as the uncertainty.

(8) There is some confusion with the use of D^* and D^{**} . I don't understand how blue and orange lines in Fig. 8(a) and (b) are straight, shouldn't the boundaries be $\nu \sim 1/Q$ as per definition of D . Also, not sure if you should quote ν^{**}/ν^* - you means some ratio of prefactors only. In Fig.8 you use D and ν/Q interchangeably - you should be more careful. In caption to the figure you refer to a disk diameter - what disk?

The reviewer is correct that lines of constant D_{visc} should look like $\eta \sim 1/Q$. However, these curves are indeed linear with slope -1 on log-log axes, as used in Figs. 8a and b. We now refer explicitly in the text to the log-log nature of this plot: "*Figure 8a shows an experimental $\eta_{\text{inv}}-Q$ phase diagram in a log-log plot.*"

To avoid confusion, we have now removed the quantities η^* and η^{**} in favour of D^* and D^{**} : "*We indicate these transitional values [D_{visc}^* and D_{visc}^{**}] in the $\eta-Q$ phase diagrams (diagonal blue and orange lines, respectively in Figs. 8a and b) for fixed $\phi=0.49$, $b=0.9\text{mm}$, and $r_{\text{out}}=13.4\text{cm}$, for which $D_{\text{visc}}^{**}=860$.*"

We have replaced the term 'disc diameter' with the outer radius r_{out} , which is the radial system size; this quantity is used in various places throughout the text.

(9) S_{front} is not clearly defined - what is the "outer perimeter of the connected advancing front"? Given that the front is still fingered, that ought to be defined somewhere.

We have rephrased the relevant paragraph as follows: "*We define a 'front instability number' $s=S_{\text{front}}/S_{\text{finger}}$. Here, S_{front} traces the interface between the outer undisturbed mass*

and the advancing pattern excluding internal structure (see red line in Figure 8a). S_{finger} is the longer internal perimeter of the finger pattern tracing the liquid-air interface. The value of s is close to 1 for a single finger where the outer front perimeter follows the internal finger interface, decreasing as fingers increasingly meet to form a common front (Fig. 8e)."

Reviewer #3 (Remarks to the Author):

The new manuscript has removed the viscously unstable situation from the previous revision and has included additional explanation for the viscously stable regime, this has led to a version of the manuscript that contextualizes the work and explains the effect more clearly. The authors have addressed concerns and comments in their response well and I recommend this manuscript for publication.

One minor comment: for the inset of Figure 5b there are no axes values. I appreciate that this is just to demonstrate the exponential relationship of the data, but I think there should still be values on the axes to make the relationship clear.

We thank the reviewer for their positive assessment, and for this suggestion. We have now elevated the previous inset of Fig. 5b to a new panel 5c. In doing so, we have included more data and added error bars and axis labels.

REVIEWERS' COMMENTS

Reviewer #1 (Remarks to the Author):

As far as I can tell the authors have clarified the paper where I had concerns and questions. It would have been much easier and less time consuming if the revised paper had indicated (say via color) where edits had been made. In one place I still find the writing ambiguous, e.g., on p. 11: "The simple model captures the overall trend in the data, despite containing none of the geometrical complexity of the branching patterns. Note, however, that there is considerable variability in the data." - The authors want to claim success in the first sentence but then admit (as I read the paper) in the second sentence that, in fact, the model does not work so well. The problem is complicated but I can appreciate that the authors have added new insights, using both experiments and simulations. I believe readers of Nature Communications will find the work valuable and interesting.

Reviewer #2 (Remarks to the Author):

I am satisfied with the changes made by to the manuscript and I am happy to recommend its publication. It mostly reads well, although I have stumbled in a few places, see below:

page 12, sentence "The fluid pressure here...": I propose changing this into "In Fig. 6b, that show data at $D_{\text{visc}}=33$, the fluid pressure (at the central node) is compared to the fluid pressure at the inlet measured experimentally."

At the end of the same paragraph you acknowledge that there is some disparity in pressured obtained experimentally and in the model, it would be good to include a reason for it (even if it is speculative). Also, I think that the fact that the number of fingers in numerics is smaller than in the experiments is consistent with that observation.

page 15, Figure 8: When you refer to this figure in the text, you acknowledge that the range of Q in a) and b) is not the same; could you please immediately stress that blue and orange lines are the same despite this?

page 16: There is ambiguity in how S_{front} is defined. What does "traces interface between the outer undisturbed mass and the advancing pattern excluding internal structure" mean. The fingers in the spoke pattern do not quite touch each other, or do their compaction fronts touch? How did you decide where to measure the front between the fingers?

We thank the reviewers for their time and constructive feedback. In the responses below, the comments of the reviewers are included verbatim in black and our replies follow in blue.

Reviewer #1 (Remarks to the Author):

As far as I can tell the authors have clarified the paper where I had concerns and questions. It would have been much easier and less time consuming if the revised paper had indicated (say via color) where edits had been made.

In fact, we did generate a separate pdf with tracked changes. We apologise if this was unintentionally omitted from our previous submission.

In one place I still find the writing ambiguous, e.g., on p. 11: "The simple model captures the overall trend in the data, despite containing none of the geometrical complexity of the branching patterns. Note, however, that there is considerable variability in the data." - The authors want to claim success in the first sentence but then admit (as I read the paper) in the second sentence that, in fact, the model does not work so well.

We disagree that there is anything ambiguous or inconsistent about this statement. Figures 3 and 5 show that the finger width $2R$ and the number of fingers N do indeed exhibit considerable variability at given values of φ and D_{visc} , which means that repeating the same experiment (or simulation) does not lead to an identical result. This is a standard feature of unstable pattern-forming systems, where instabilities grow exponentially from small, random perturbations. However, both $2R$ and N do exhibit clear trends with φ and D_{visc} , and those trends are well captured by our simple models. Thus, we stand by our results and by this paragraph in the manuscript.

The problem is complicated but I can appreciate that the authors have added new insights, using both experiments and simulations. I believe readers of Nature Communications will find the work valuable and interesting.

We thank the reviewer for this positive assessment and for all of their previous constructive feedback, which has helped us to greatly improve the paper.

Reviewer #2 (Remarks to the Author):

I am satisfied with the changes made by to the manuscript and I am happy to recommend its publication.

We thank the reviewer for their positive recommendation and for their constructive comments in previous rounds, which helped us to greatly improve the manuscript.

It mostly reads well, although I have stumbled in a few places, see below:

page 12, sentence "The fluid pressure here...": I propose changing this into "In Fig. 6b, that show data at $D_{\text{visc}}=33$, the fluid pressure (at the central node) is compared to the fluid pressure at the inlet measured experimentally."

We agree with the reviewer that this sentence was awkward. We have adopted a slightly revised version of the reviewer's suggestion: "Figure~6b shows the evolution of injection pressure for $D_{\text{visc}}=31$, comparing the experimental pressure measured at the inlet with the simulation pressure calculated at the central node for three realisations." We have also revised the surrounding text to address the next comment.

At the end of the same paragraph you acknowledge that there is some disparity in pressured obtained experimentally and in the model, it would be good to include a reason for it (even if it is speculative). Also, I think that the fact that the number of fingers in numerics is smaller than in the experiments is consistent with that observation.

We thank the reviewer for the suggestion. We have now revised this paragraph and the following one to better compare and contrast the experiments. Much of this discussion has been relocated from what was previously the last paragraph of the Methods section. The two paragraphs now read:

"Figure 6a shows a detailed view of a simulation at moderate $D_{\text{visc}}=22$ with multiple active fingers. The skeletonized viscous flow network shows the fluid pressure, which decreases toward the tips of active fingers and is uniform along inactive fingers, with the maximum pressure at the inlet. Figure 6b shows the evolution of injection pressure for $D_{\text{visc}}=31$, comparing the experimental pressure measured at the inlet with the simulation pressure calculated at the central node for three realisations. In both cases, the pressure initially increases rapidly as bulldozing mobilises friction and capillary forces, then more slowly due to build-up of viscous pressure as fingers grow in length. The simulations reproduce the trend in the pressure data from experiments, as well as the typical frequency and magnitude of the fluctuations; however, the overall magnitude of the injection pressure is somewhat higher in the simulations than in the experiment. Figure 2b shows simulation results across a range of filling fractions and injection rates, reproducing the experimental transition from single-finger to multi-finger growth as a function of Q , although somewhat under-predicting the number of fingers at high Q . Figure 3 includes the simulated finger width $2R$ at low Q , which decreases a bit more steeply with φ than what is observed in the experiments.

These differences between experiment and simulation may be due to the fact that the simulations use an exponential friction model (Eq. 11) along the entire the compaction front, regardless of curvature. Our experimental data indicates that friction along the finger side walls is indeed well captured with an exponential model (Fig. 5), but that friction at the finger tips is better captured by a linear friction model (Fig. 3). We have used the exponential model in the simulations to prioritise viscous stabilisation, which is controlled by side-wall friction. However, tip friction controls the finger width and the resistance to finger propagation, so it is not surprising that the simulations exhibit a steeper variation of $2R$ with φ and a larger injection pressure than observed in experiments. Resolving this discrepancy would require the use of curvature-dependent friction along the compaction front, which may be the subject of future work."

page 15, Figure 8: When you refer to this figure in the text, you acknowledge that the range of Q in a) and b) is not the same; could you please immediately stress that blue and orange lines are the same despite this?

We thank the reviewer for the suggestion. We have added the following explanation: “*Note that the D_{visc}^* and D_{visc}^{**} lines (blue and orange, respectively) appear steeper in Fig. 8b than in Fig. 8a because the range of Q is wider in the former.*”

This sentence is added in the paragraph where the boundary lines are first introduced (pg. 16).

page 16: There is ambiguity in how S_{front} is defined. What does “traces interface between the outer undisturbed mass and the advancing pattern excluding internal structure” mean. The fingers in the spoke pattern do not quite touch each other, or do their compaction fronts touch? How did you decide where to measure the front between the fingers?

We have rephrased the description to clarify this point: “*We define a ‘front instability number’ $s = S_{\mathrm{front}}/S_{\mathrm{finger}}$, where S_{front} is the length of outer edge of the compaction front (i.e., the outer boundary between the compaction front and the undisturbed material; see red contour in Figure 8c). For the spoke pattern, this boundary becomes nearly circular since the compaction fronts of neighbouring fingers touch.*”